# Machine learning-assisted wearable sensing systems for speech recognition and interaction

Tao Liu, Mingyang Zhang, Zhihao Li, Hanjie Dou, Wangyang Zhang, Jiaqian Yang, Pengfan Wu, Dongxiao Li ⊠ & Xiaojing Mu ⊠

The human voice stands out for its rich information transmission capabilities. However, voice communication is susceptible to interference from noisy environments and obstacles. Here, we propose a wearable wireless flexible skin-attached acoustic sensor (SAAS) capable of capturing the vibrations of vocal organs and skin movements, thereby enabling voice recognition and human-machine interaction (HMI) in harsh acoustic environments. This system utilizes a piezoelectric micromachined ultrasonic transducers (PMUT), which feature high sensitivity (-198 dB), wide bandwidth (10 Hz-20 kHz), and excellent flatness (±0.5 dB). Flexible packaging enhances comfort and adaptability during wear, while integration with the Residual Network (ResNet) architecture significantly improves the classification of laryngeal speech features, achieving an accuracy exceeding 96%. Furthermore, we also demonstrated SAAS's data collection and intelligent classification capabilities in multiple HMI scenarios. Finally, the speech recognition system was able to recognize everyday sentences spoken by participants with an accuracy of 99.8% through a deep learning model. With advantages including a simple fabrication process, stable performance, easy integration, and low cost, SAAS presents a compelling solution for applications in voice control, HMI, and wearable electronics.

The human body generates a wealth of biological signals that can be detected, digitized, analyzed, and interacted with external devices[1-3]. Among these, the human voice is particularly notable for its rich information transmission capabilities in the time, frequency, and amplitude domains[4]. This rich information-carrying capacity makes sound a critical component in bio-communication, human-machine interaction (HMI), and Internet of Things (IoT) applications, including smart homes, remote control, identity recognition, and voice-based systems. However, voice communication based on air vibration is susceptible to interference and obstruction by background noise (such as roadsides, shopping malls, stations, and other noisy environments) and acoustic media (such as in fires, hospitals, underwater, and other

special scenes)[5,6]. Moreover, the vocalization process relies on a coordinated system of organs, and any damage - due to conditions such as amyotrophic lateral sclerosis (ALS), stroke, Parkinson's disease, or laryngeal cancer - can severely impact voice clarity and recognition efficiency[7,8]. To address these challenges, researchers have developed advanced noise reduction algorithms and multi-microphone systems to enhance voice processing capabilities. However, the effectiveness of these solutions is constrained by the quality of the sound signals and the complexity of multi-feature parameters[9,10]. For instance, single-microphone systems fail to capture spatial features and struggle to deliver audio signals with a high signal-to-noise ratio[11]. While multi-microphone systems and associated algorithms improve speech signal

Key Laboratory of Optoelectronic Technology & Systems of Ministry of Education, International R & D Center of Micro-nano Systems and New Materials Technology, Chongqing University, Chongqing 400044, China. ⊠e-mail: lidongxiao@cqu.edu.cn; mxjacj@cqu.edu.cn

processing, they require intricate engineering designs and occupy more space[12,13].

Recently, visual speech recognition based on facial and lip movements has emerged as a method for enhancing speech perception in noisy environments[14–16]. While this approach improves speech perception quality in challenging acoustic conditions, it requires additional cameras, which increase system complexity and reduce practicality. In recent years, attachable sensors that directly monitor facial motion states have gained attention as a solution for silent speech recognition[17–25]. Although facial features can complement audio signals to some extent, they have significant limitations in capturing acoustic parameters such as pitch, timbre, and sound intensity. Conversely, placing sensors directly on the vocal organ area offers an effective way to achieve comprehensive sound information collection[26–29]. Traditional wearable devices for monitoring sound signals are typically attached to the body using straps or adhesive patches[30–33]. However, their rigidity and flat shape restrict practical applications. The development of flexible materials and sensing technologies has paved the way for imperceptible skin-wearable devices[34–36]. Currently, the flexible sensing technologies installed in vocal organs mainly include graphene, flexible surface electromyography electrodes, piezoresistive and triboelectric[26,37–44]. Compared to conventional rigid microphones, these technologies are more comfortable to wear and integrate seamlessly into daily activities. Despite their advantages, these sensors often rely on wired hardware, limiting their adaptability for routine use.

To overcome these challenges, integrated flexible wearable devices[45–48] with signal processing and transmission units[49,50] is crucial to fully utilize the potential of various electromechanical features[51–54]. Advances in microelectromechanical systems (MEMS) manufacturing technology have brought hope for improving the integration of wearable devices. A notable innovation is the incorporation of commercial MEMS accelerometer chips into wearable devices, enabling continuous monitoring of mechanical sound signals, such as speech, swallowing, breathing, and cardiac movements[4,27,55]. However, current sensors fail to meet requirements for wide frequency band ranges and flatness, limiting the energy distribution of signal spectrums. Furthermore, detecting skin acceleration provides only muscle movement pattern data, neglecting critical vibration information from vocal organs. This lack of biometric information results in relatively weak mechanical sound signals when monitoring small amplitude muscle movements. This limitation is particularly unfriendly for users with thicker skin tissue (such as those with thyroid enlargement) or laryngeal injuries. Therefore, it is necessary to develop a new portable voice interaction system to address these issues and advance user experience and HMI.

In this study, we demonstrated a wearable skin-attached acoustic sensor (SAAS) designed for speech recognition and HMI in noisy environments. The piezoelectric micromachined ultrasonic transducers (PMUT) is the core sensing component of the system, which has the characteristics of small footprint, high sensitivity, wide bandwidth and excellent flatness. To enhance the integration and wearing comfort of the sensing system, we used soft electronic technology and elastomer packaging technology. In the device packaging, we used spatial isolation and serpentine conductive wires to mechanically separate the sensing element from the supporting electronic device to minimize the interference of circuit noise and improve the measurement sensitivity. The developed SAAS exhibits high sensitivity to both vocal organ muscle movements and vocal cord vibrations, enabling it to capture high-quality signals across a wide frequency range of 10 Hz to 20 kHz while effectively reducing environmental noise interference. In demonstrations comparing SAAS with commercial microphones, the system remained largely unaffected by harsh acoustic environments and masks. Furthermore, the robustness and flexibility of SAAS were validated by testing under increased head movement and varying

neck placement of the sensing system. These characteristics allow SAAS to eliminate the need for precise placement and significantly reduce motion artifact interference in acoustic signal capture. Based on the rich biological feature information obtained by SAAS and combined with artificial intelligence, we achieved 99.5%, 100% and 96.9% accuracy rate in tasks such as phonemes, tones, and words with similar pronunciations classification, respectively. This exceptional performance underscores its potential for identity recognition and security systems based on wearable acoustic devices. Additionally, we demonstrated the advantages of wearable acoustic devices in human-machine interaction and IoT control. Finally, a dataset of ten sentences from participants' daily lives was collected and classified using deep learning with an accuracy of 99.8%. These results indicate that the highly integrated wearable acoustic sensor can not only effectively perform speech recognition in harsh acoustic environments, but also show great potential in speech interaction applications involving patients with speech disorders.

## Results

### Sensing mechanism of SAAS

The sensing mechanism and key functions of SAAS are illustrated in Fig. 1. As shown in Fig. 1a, the information flow of SAAS in HMI is demonstrated. For instance, when participants intend to control a robot dog to transition from standing on all fours to an upright posture, they use their vocal organs to generate the corresponding voice commands. However, in a noisy environment, these vocal commands cannot be directly transmitted to the robot dog through the air due to interference. To address this limitation, we have developed a sensor that can be directly attached to the throat. This sensor can capture the acoustic signals generated by the vocal organs, even in noisy environments. The sensor then transmits the collected voice commands to the robot dog via an integrated signal processing circuit and wireless module. The robot dog is equipped with a voice recognition module that analyzes the received commands and executes the corresponding actions. It is worth emphasizing that the sensor depicted in Fig. 1a possesses several notable characteristics, including a small footprint, softness, skin-friendliness, and robustness. These features enable the SAAS to adhere closely to the skin, ensuring stable operation even during daily activities.

The exploded view in Fig. 1b provides a detailed visualization of the overall structure of the sensing system. The system comprises a flexible printed circuit board (FPCB), electronic components, PMUT, a bluetooth low energy (BLE) module, a rechargeable battery, and a flexible silicone encapsulation layer (see Supplementary Fig. S1 for the physical decomposition of the sensor). The FPCB employs a 100 μm thick polyimide layer as the supporting substrate. The top and bottom surfaces of the substrate layer are patterned with 12 μm thick rolled annealed copper (Cu), which is encapsulated with a 50 μm thick polyimide insulating layer on each side. Various functional electronic components are integrated into the FPCB. Despite the presence of rigid electronic components, the deformable FPCB substrate, serpentine interconnects, and flexible encapsulation materials enable the entire system to exhibit low-modulus elastic mechanics. The PMUT, which serves as a critical component for speech recognition, is highlighted in Fig. 1c, where its exploded view provides further insight into its structure and functionality. Compared to traditional microphones (such as dynamic microphones or electret microphones), the PMUT offers several advantages, including a smaller footprint ($3.5 \times 3.5$ mm²), lighter weight (2.8 mg) and a wider frequency response range (10 Hz–20 kHz). Further details on the PMUT will be discussed in the next section. The BLE module (ESP32) enables high-speed wireless data transmission, and the entire sensing system is powered by a rechargeable lithium-ion polymer battery (60 mAh).

To achieve softer and more skin-compatible mechanical properties, the sensing system adopts a distributed layout scheme.

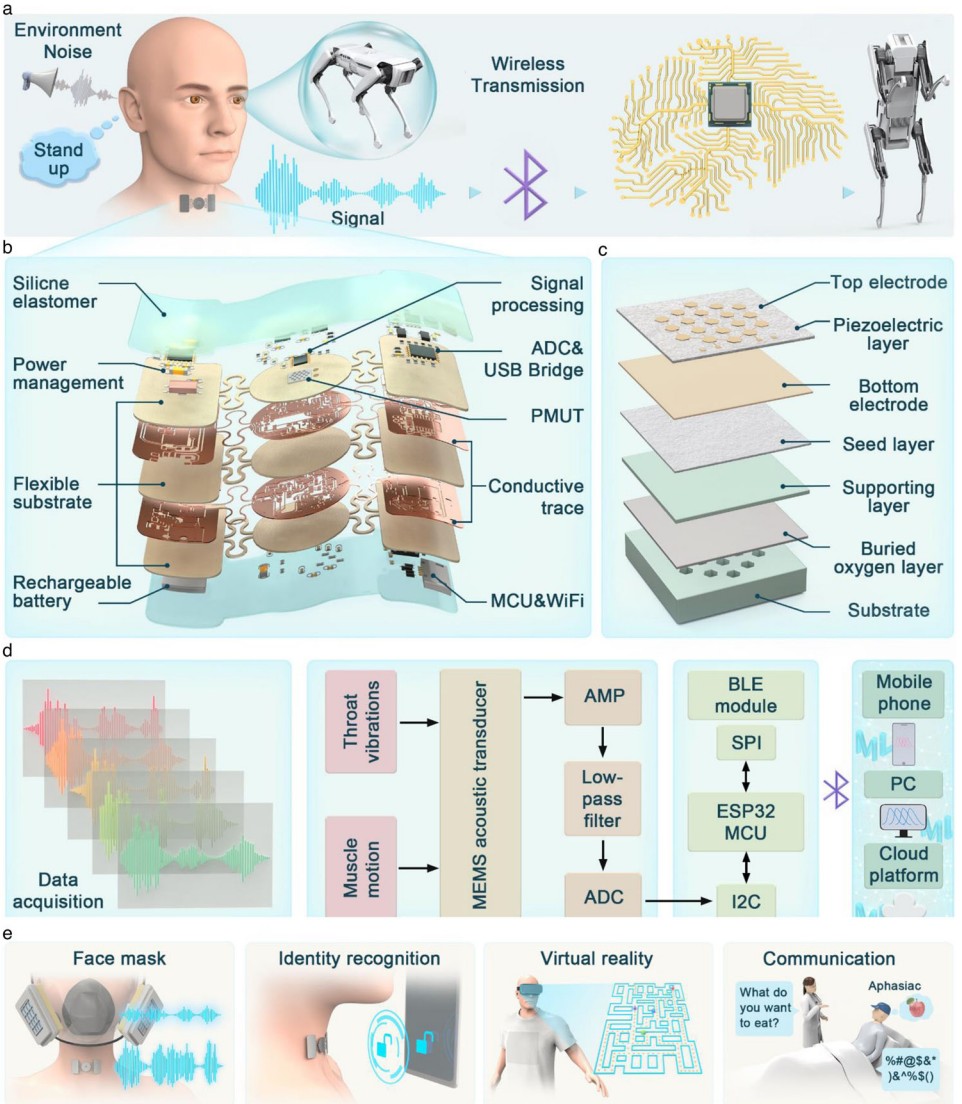

**Fig. 1 | Wireless, flexible, attachable acoustic sensor for speech recognition in harsh acoustic environments. a** Schematic diagram of the speech recognition system to achieve HMI. **b** Exploded view of the attachable acoustic sensing system. **c** Structural scheme of PMUT. **d** Flowchart of the steps for processing the vibration and muscle movement signals of the vocal organ, including signal processing, control, wireless communication, and display terminal. **e** Schematic diagram of the application of SAAS in speech recognition and interaction.

Specifically, mechanical components and peripheral signal processing circuits are arranged in th"e central circular area of the FPCB, the battery and power management circuit are positioned in the left area, and the microprocessor and wireless transmission module are located in the right area (See Supplementary Fig. S2 for circuit details). This distributed layout not only enhances skin compatibility but also effectively reduces circuit noise interference, resulting in a signal-to-noise ratio (SNR) of 32.64 dB (Supplementary Fig. S3). The different areas of the sensing system are interconnected by serpentine traces, which provide greater deformation freedom. This flexible interconnect structure enables the sensing system to conform closely to the throat's curves and movements, improving both wearing comfort and sensor durability. SAAS is attached to the skin using a 3 M 9907 T biomedical adhesive. This flexible interface minimizes irritation or discomfort, even when applied to sensitive areas such as the laryngeal process (Supplementary Fig. S4).

Figure 1d illustrates the overall workflow of SAAS, including signal acquisition, preprocessing, remote transmission, and deep learning. First, the MEMS acoustic transducer positioned on the throat captures the mixed acoustic signals, including throat vibration and muscle movement generated by the vocal organs. These collected acoustic signals are preprocessed using a filter-amplifier circuit and subsequently converted into digital signals by an analog-to-digital converter. The digital signals are then transmitted in real-time to a terminal device (mobile phone, personal computer or cloud) via the BLE module for further processing. A denoising algorithm filter out redundant noise, and the signals are classified and interpreted using a neural network model. Finally, a custom terminal application is designed with a graphical user interface that records, stores, and displays relevant voice data in real-time. Overall, the integration of advanced technologies - including micro-nano sensing, system integration, machine learning, and flexible packaging - offers a promising solution for enabling multi-task voice interaction scenarios (Fig. 1e).

## Design principle and characterization of the device

The PMUT, serving as the receiving sensor in SAAS, converts the throat's acoustic signals into electrical signals via the piezoelectric effect. A schematic of a single PMUT is shown in Fig. 2a. The PMUT

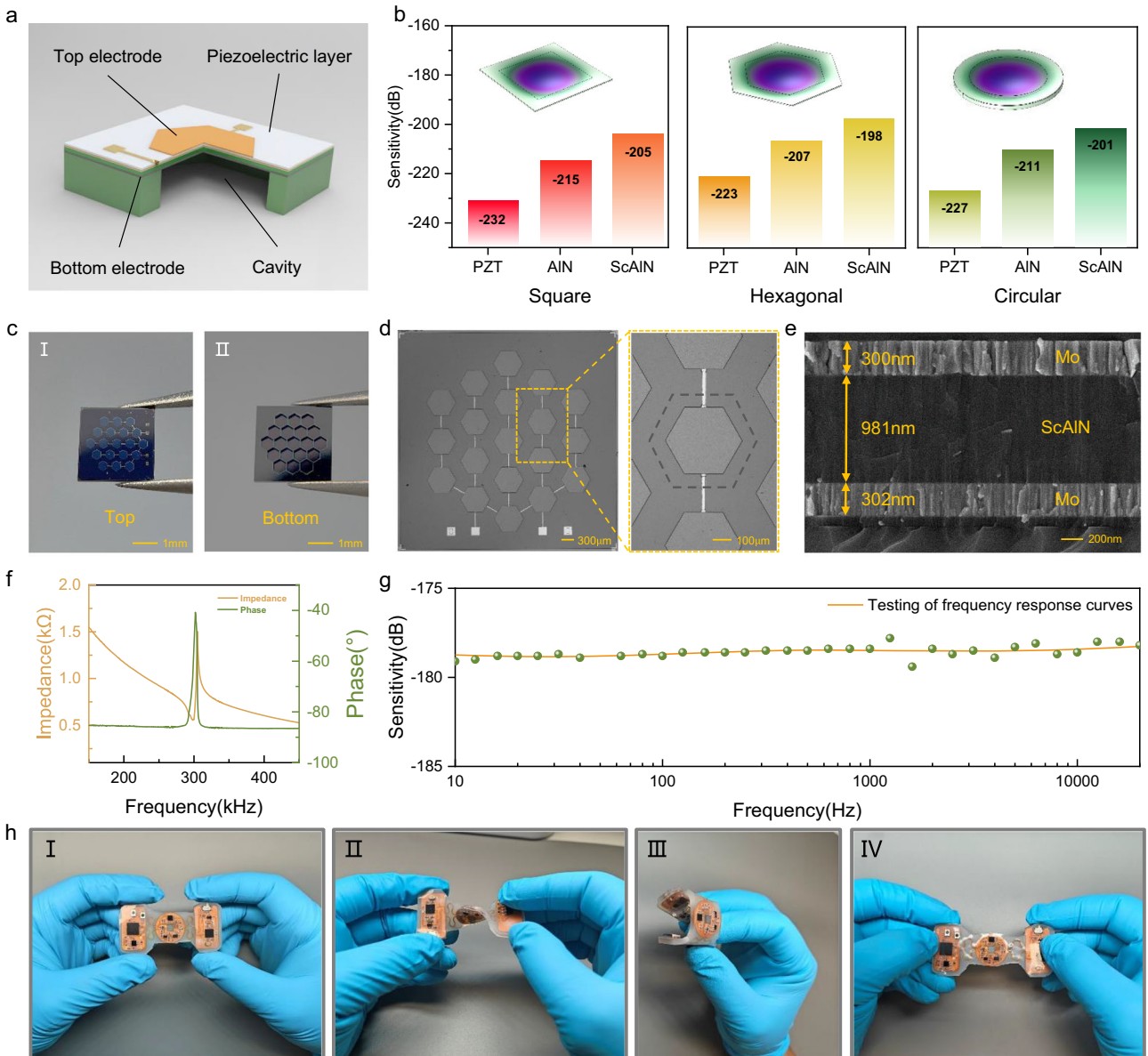

**Fig. 2 | Design principle and characterization of the device. a** Cross-sectional view of the three-dimensional structure of the acoustic sensor based on SOI wafer. **b** Finite element simulation comparison of relevant parameters of acoustic sensors of PZT, AlN, ScAlN and the corresponding three shapes. **c** Photos of the front (I) and back (II) of the PMUT. **d** Optical microscope image of the prepared acoustic chip and close-up details of a microelement. **e** SEM cross-sectional view of the Mo/ ScAlN/Mo film structure of the prepared acoustic chip. **f** Resonant frequency response of the electrical impedance magnitude and phase of the MEMS chip in air. **g** Sensitivity test curve of the packaged MEMS sensor in the low-frequency wide-band range in water. **h** Images display of the flexible device under undeformed (I), twisted (II), bent (III) and stretched (IV).

primarily consists of a top electrode, a piezoelectric film, a bottom electrode and a cavity (see Supplementary Table S1 for detailed parameter settings)[56,57]. Here, the area covered by the electrode generates a stress along the thickness of the film, which forces the film to produce a bending moment about the neutral axis. The cavity provides the necessary vibration conditions for the film (Supplementary Fig. S5a)[58,59]. In speech recognition applications, the key characteristics required for devices are high sensitivity, wide bandwidth and excellent flatness. To develop high-performance sensing devices, we optimized both the materials−lead zirconate titanate (PZT), aluminum nitride (AlN), 20% scandium-doped aluminum nitride (ScAlN)−and the geometrical structures−square, circular, and hexagonal−of PMUT using simulation (see Methods and Supplementary Fig. S6 for details). We designed the resonant frequency of the PMUT to be 300 kHz

(Supplementary Fig. S6a), ensuring flat sensor sensitivity in the low-frequency range. This approach significantly reduces the sensor's volume compared to sensors operating at their resonant point. Figure S6b illustrates the receiving sensitivity of various PMUT designs within the 10 Hz−20 kHz frequency band. Generally, for a given structural design and driving voltage, the receiving sensitivity of the PMUT is proportional to the ratio of the piezoelectric constant ($d_{33}$) to the dielectric constant ($\varepsilon_r$) of the piezoelectric material[60,61]. Since ScAlN exhibits a higher $d_{33}/\varepsilon_r$ ratio (Supplementary Table S2), ScAlN-based PMUTs demonstrate superior reception sensitivity (Fig. 2b). Next, we discuss the impact of PMUT structure on sensitivity. With consistent resonant frequency, hexagonal designs typically demonstrate higher sensitivity compared to square and circular designs (Fig. 2b). Moreover, hexagonal structures offer a better filling ratio

than the circular designs, enhancing area utilization in array arrangements. Due to the advantages of the ScAlN combined with the hexagonal structure in terms of sensitivity (−198 dB), bandwidth (10 Hz–20 kHz), and flatness (±0.5 dB), the optimized design improved the device performance. To further enhance receiving sensitivity, multiple PMUT units were electrically paralleled to accumulate more charge during operation. Meanwhile, this array design also increases the contact area with the skin, thereby collecting more acoustic signals to improve the performance and reliability of voice recognition.

Next, we completed the PMUT fabrication using standard MEMS processes (see Methods and Supplementary Fig. S7 for fabrication details). Figure 2c shows the top view and back view of the PMUT, and the designed sensor array size is $3.5 \times 3.5$ mm² (see Supplementary Fig. S5b for detail size of PMUT element). Figure 2d displays the optical microscope image of the PMUT array and a magnified view of a single PMUT, while Fig. 2e presents the cross-sectional scanning electron microscope (SEM) image of the fabricated Mo/ScAlN/Mo stack, clearly showing the boundaries between layer. The sputtered ScAlN film exhibits excellent crystal orientation. Molybdenum (Mo) was selected for the top and bottom electrodes due to its good lattice match with ScAlN. Energy-dispersive X-ray spectroscopy (EDS) analysis confirmed the uniform distribution of Sc, Al, and N elements in the piezoelectric layer (Supplementary Fig. S8). X-ray diffraction (XRD) measurements revealed that the (002) peak of the ScAlN film appears near 36° (Supplementary Fig. S9). Impedance analysis results corroborated simulation results (Supplementary Fig. S6a), indicating a resonant frequency of 302 kHz for the PMUT sensor (Fig. 2f). This resonant frequency ensures both high sensitivity (−178 dB) and excellent flatness (±0.5 dB) within the operational bandwidth of 10 Hz–20 kHz (Fig. 2g). Silicone (Ecoflex-0030) was used to encapsulate the densely packed electronic components, providing a flexible and robust design. Figure 2h shows images of the encapsulated device under various mechanical deformations, including un-deformed (I), twisted (II), bent (III) and stretched (IV). Meanwhile, Supplementary Fig. S10 shows the robust mechanical properties of the SAAS under various mechanical deformations. In addition, we tested the structural stability of the SAAS with a shaker (Supplementary Fig. S11). These results confirm the SAAS's stability, low-modulus, and elastic response, making it well-suited for wearable applications.

## Anti-interference voice recognition

In noisy environments such as firefighting or quarantine scenarios, reliable sound transmission is essential for effective communication[62]. However, commercial microphones often struggle to recognize sounds accurately in harsh acoustic conditions or when the user is wearing a mask[63]. To evaluate signal detection performance under background noise and acoustic medium interference, we compared the SAAS with a commercial reference microphone. The test system comprised a SAAS, a commercial microphone, an audio source, and an oscilloscope (Supplementary Fig. S12). The commercial microphone (CRY333) was positioned 30 cm in front of the tester's mouth. Participants vocalized the letters "CQU", and both SAAS and the commercial microphone recorded the sound information simultaneously. The recorded sound was converted into electrical signals and displayed on dual channels oscilloscope. The time domain waveform curves were then used to construct spectrogram through short-time Fourier transform (STFT), providing an intuitive visualization of the frequency range and signal intensity.

First, we attached the SAAS to the participant's throat and conducted tests in a quiet environment (40 dB) (Fig. 3aI). Both the SAAS and commercial microphone recorded time domain signals within the (0–1 kHz) frequency band (Fig. 3bI, cI). The signals from both devices showed three distinct segments, with the time interval between "C" and "Q" being greater than that between "Q" and "U". The recorded signals from both methods were highly consistent, demonstrating that

the SAAS provides excellent recording accuracy comparable to commercial microphones under quiet conditions. Next, the performance of the SAAS in a noisy environment was verified, and commercial audio was selected as a controllable noise source to simulate the noise environment (85 dB). Among them, the frequency range and intensity of the noise are larger than the participants' voice (Fig. 3aII). When vocalizations were recorded under these conditions, the SAAS effectively identified vocal organ movement information. The time domain waveform and spectrogram of the signal (Fig. 3bII) are basically consistent with those in a quiet environment (Fig. 3bI). In contrast, the commercial microphone failed to distinguish human voices from environmental noise accurately, as shown in Fig. 3cII. In addition, we tested the noise immunity performance of the device in higher decibel noise environments (including 95 dB, 105 dB, 115 dB, and 125 dB) (Supplementary Fig. S13). The experimental results fully demonstrate the excellent anti-interference ability of SAAS in the noisy environment. To test performance under acoustic medium interference, participants wore gas masks that covered their faces, affecting sound transmission. Under these conditions (Fig. 3aIII), the SAAS continued to identify signal patterns effectively, with test results (Fig. 3bIII) closely matching those from quiet conditions (Fig. 3bI). Conversely, the commercial microphone suffered significant degradation, losing high-frequency band signals and exhibiting a reduced overall amplitude (Fig. 3cIII). Overall, these results demonstrate that the SAAS effectively mitigates the effects of noisy environments and acoustic media on sound transmission. This capability makes the SAAS highly suitable for critical applications such as firefighting, epidemic prevention, underwater communication, and sound transmission in noisy environments.

To verify the performance of the SAAS and demonstrate its signal pattern recognition capability and usage compatibility under varying conditions, we used the sensor attached to the throat to record the tester's vocalization of the word "perfect". First, we examined the effect of different attachment positions on signal characteristics by attaching the sensor near the throat in positions above, below, left, and right of the laryngeal center (Fig. 3d). This experiment allowed us to assess how positional adjustments influence signal quality and recognition accuracy. Second, we investigated the effect of neck movements —raising the head, lowering the head, and turning the head to the right or left (Fig. 3d)—on signal recognition. This assessment aimed to determine whether subtle local muscle changes interfere with the SAAS's signal recognition capabilities. The SAAS collected the time domain curves of signals (Fig. 3e) from the participants across nine different attachment positions or neck movements. The results revealed that when the device was attached to the center of the larynx, the signal amplitude was the highest, suggesting this was the optimal position for testing (Supplementary Fig. S14). It is worth noting that the signals collected in the nine usage conditions have highly consistent characteristics (Supplementary Fig. S15). To further verify the wearing stability of the device, we compared the device's attachment status and signal acquisition capabilities before and after the participants ran (Supplementary Fig. S16). In summary, these experiments highlight the SAAS's robust performance and wearing stability, making it suitable for diverse usage conditions.

## Voice recognition and identity authentication

With the rapid expansion of the global smart market, personal devices increasingly store sensitive user information, such as social data, location details, health status, and sleep patterns. Consequently, the security of personal devices has become the focus of people's attention[64,65]. Currently, the most commonly used security protocol for personal terminals is password entry. However, text passwords are vulnerable to being leaked. Similarly, voice passwords collected through commercial microphones are susceptible to interference from background noise and acoustic media. The SAAS collects the physiological mechanical acoustic signals generated by the user's

vocal organs and has the characteristics of good flexibility, skin-friendliness, and anti-interference, providing a new possibility for wearable voice-controlled security systems. To evaluate the voiceprint recognition capability of the SAAS, we conducted classification

learning tests on fundamental phonetic elements, including phonemes (Fig. 4a), tones (Fig. 4b), and words (Fig. 4c) with similar pronunciations commonly used in daily language. Measurement data (see Supplementary Figs. S17−S19) were analyzed using a convolutional neural

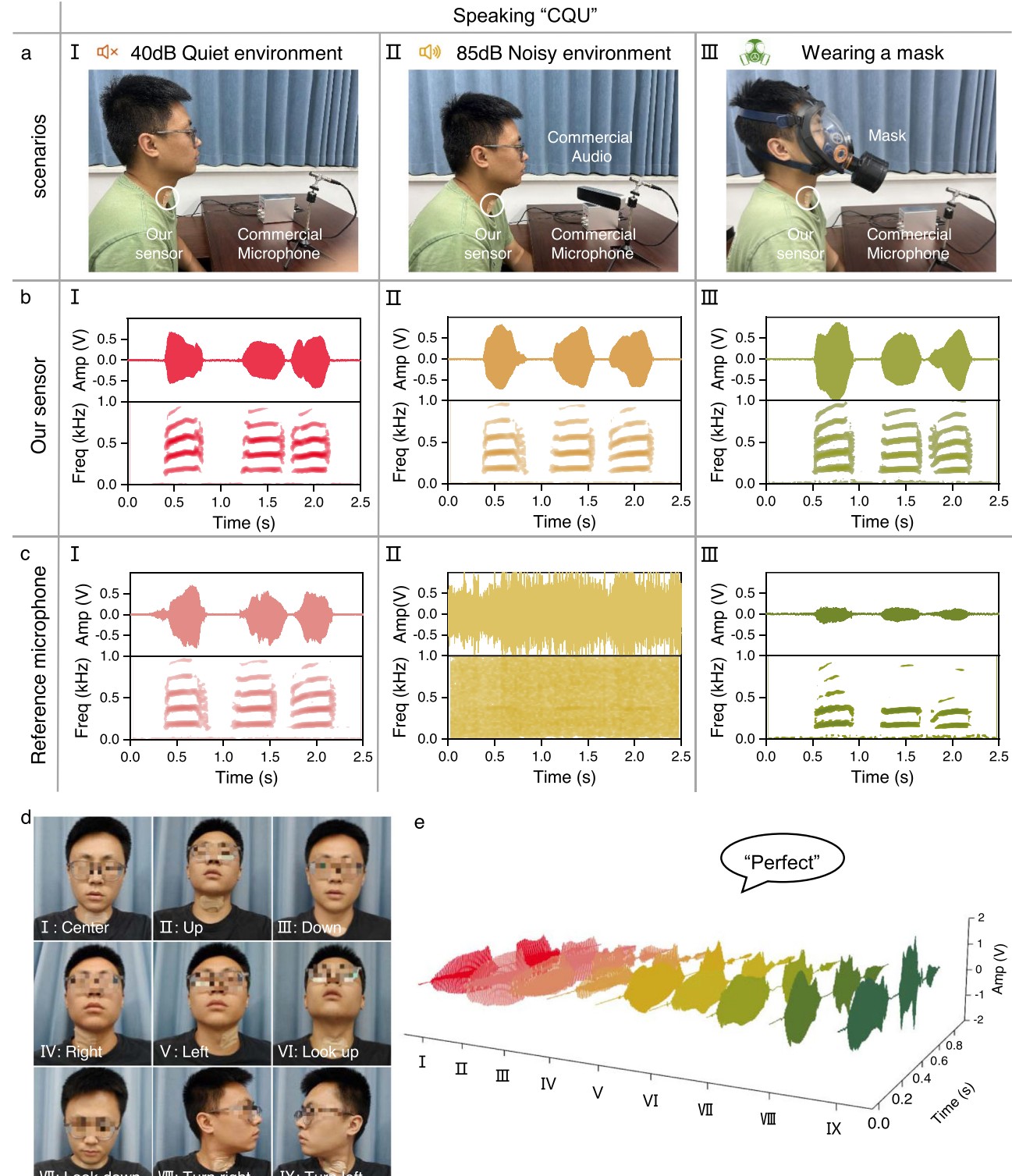

**Fig. 3 | Comparative experiments on speech detection in harsh acoustic environments. a** Photographs of the same comparison test in a quiet environment (I), a noisy environment (II), and wearing a mask (III). **b** When the participant said "CQU" in a quiet environment (I), a noisy environment (II), and wearing a mask (III), SAAS showed the time domain waveform and spectrum information of the sound signal. **c** When the participant said "CQU" in quiet environments (I), noisy

environments (II), and wearing masks (III), the commercial reference microphone showed the time domain waveform and spectrum information of the sound signal. **d** Photos of the test subjects in 9 throat attachment positions and movements. **e** Time domain waveforms obtained when saying "Perfect" through SAAS in 9 throat attachment positions and movements.

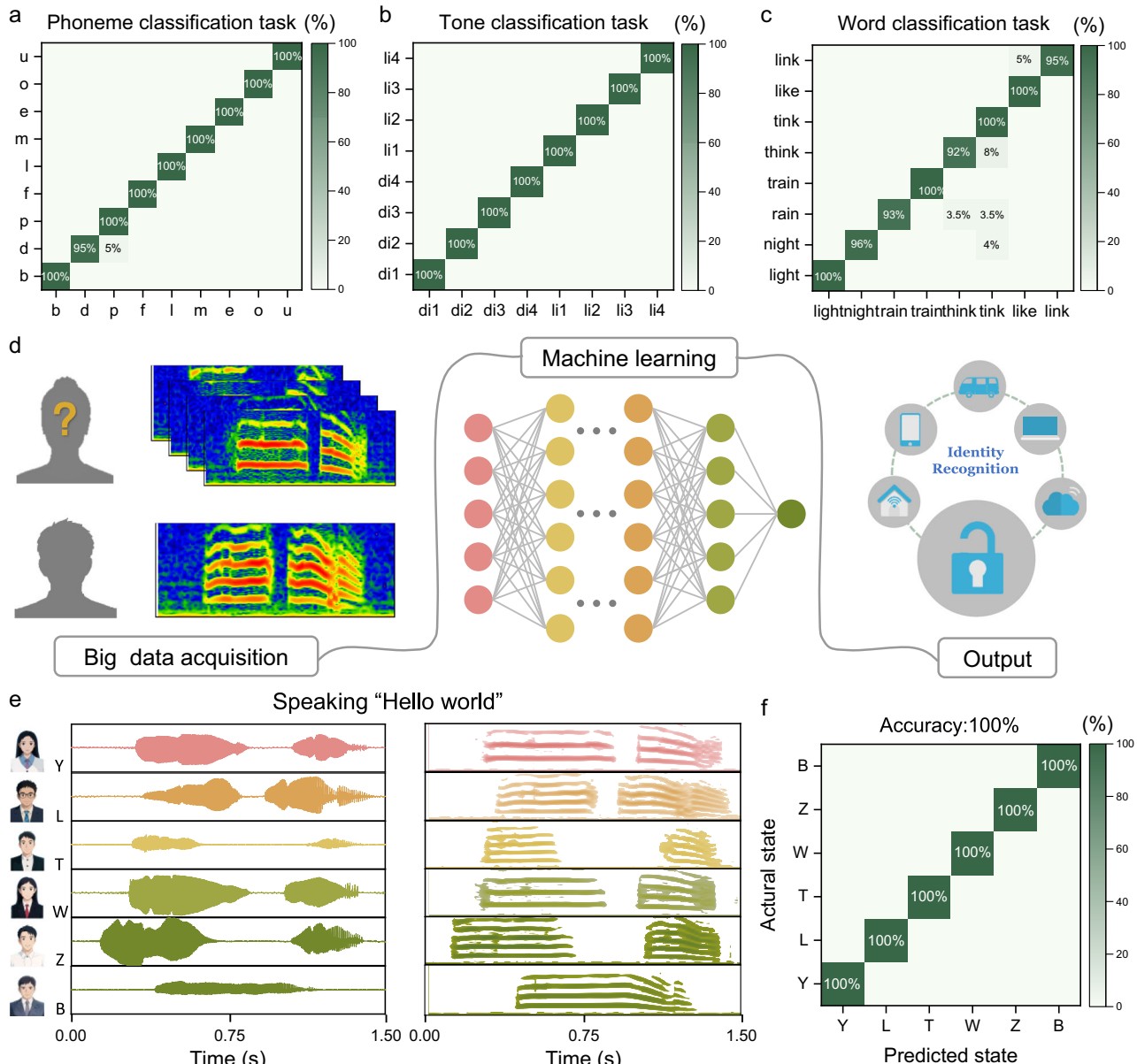

**Fig. 4 | Demonstration of identity recognition based on SAAS. a–c** Confusion matrix of the classification task of phonemes, tones, and similar-pronunciation words. **d** Schematic diagram of the identity recognition system realized through data collection, deep learning, and real-time display. **e** Acoustic information from different participants when they say "hello world". **f** Confusion matrix of identity recognition.

network model. The confusion matrix summarized the classification accuracy, and the recognition accuracy reached 99.5%, 100% and 96.9% respectively. These results underscore the effectiveness of the SAAS in distinguishing subtle acoustic features in speech.

We further demonstrated the application of wearable acoustic devices in identity authentication systems (Fig. 4d), which are widely used for personal information security in fields such as smart homes, mobile terminals, and virtual reality devices[66]. In identity authentication, we set "Hello World" as the login password. Even if an unauthorized user spoke the correct password, the system identified them as an intruder by analyzing the unique waveform characteristics of their voice using a neural network model. To validate this, the administrator "L" recorded the voice password "Hello world", which was set as the login credential. Five unauthorized users subsequently repeated the same phrase as an intrusion attempt (Fig. 4e). Using data from six participants and analyzing 123 recorded samples through filtering and deep learning, the system successfully identified the

administrator's unique voice waveform. The algorithm extracted specific features to isolate the administrator's acoustic signature from those of unauthorized users. The confusion matrix summarizing the system's performance indicated a recognition accuracy of 100% (Fig. 4f). The Supplementary Movie S1 shows the whole process of identity recognition of administrators and intruders by wearing SAAS in a noisy environment. These results highlight the SAAS's exceptional anti-interference capabilities, making it suitable for integration into other electronic devices for enhanced voice recognition in harsh acoustic environments. This adaptability opens new possibilities for the use of voice-controlled systems in fields requiring high security, even under challenging conditions.

## Human-machine interaction
The SAAS presented in this article can be applied to a wide range of HMI scenarios, including virtual reality, smart homes, resource exploration, and specialized task execution. The participants' signal

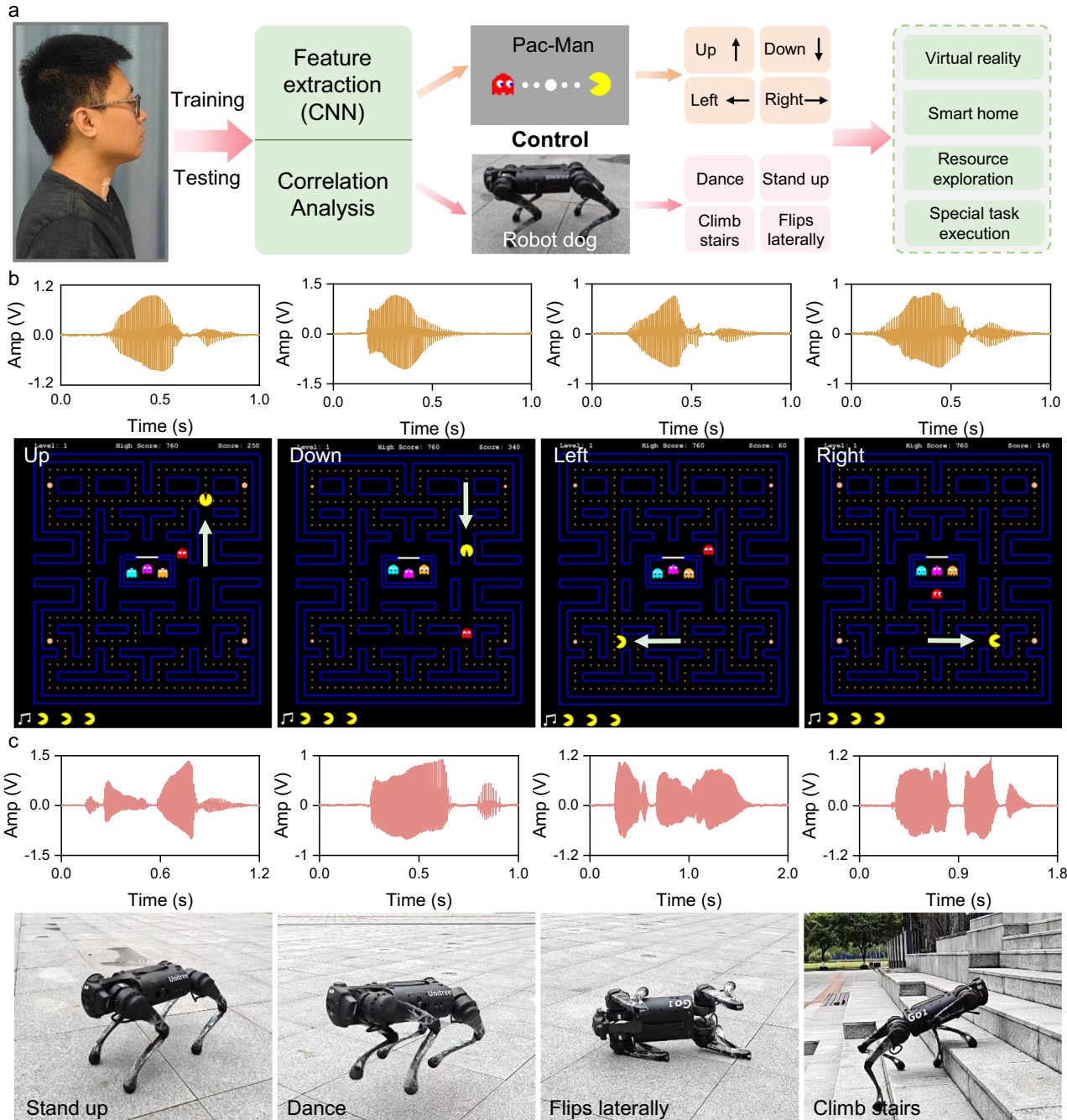

**Fig. 5 | Using SAAS to control the virtual game and the robot dog. a** Schematic diagram of the real-time wireless HMI control system. **b** Demonstration of voice commands up, down, left, and right commands in Pac-Man game. **c** Remote wireless control of the robot dog to perform actions through voice commands: "Stand up", "Dance", "Flips laterally", "Climb stairs".

instructions are collected through SAAS and feature extraction is performed. Then the signal instructions are judged in real time by the neural network model and the actions corresponding to the instructions are executed. To demonstrate the sensor's applicability in HMI, we developed a real-time wireless Pac-Man game and a robot dog control system (Fig. 5a). We illustrated the potential application of SAAS in virtual reality using the Pac-Man game as an example. Pac-Man has two degrees of freedom: moving vertically (up and down) and horizontally (left and right). As preparation, four signal instructions −"Up", "Down", "Left" and "Right"−were recorded in advance and stored in the system. These instructions were mapped to specific game movements: "Up" for forward, "Down" for backward, "Left" for turning left, and "Right" for turning right. When the operator speaks a

command, SAAS captures the vocal organ vibration signal and transmits it to the terminal system via the Bluetooth module. The signal lag of this process is about 360 ms (Supplementary Fig. 20 and Supplementary Movie S2). After the neural network model processes the data, the system recognizes the signal instructions. Simultaneously, text instructions and corresponding Pac-Man actions are displayed on the terminal. We captured the Pac-Man game screen during gameplay as the participant sequentially said: "Up", "Down", "Left" and "Right" (Fig. 5b and Supplementary Movie S3).

Next, we explored the application potential of SAAS in controlling mechanical equipment using a real-time wireless robot dog control system. Four signal commands−"Flips laterally", "Stand up", "Climb stairs" and "Dance"−were pre-recorded and stored in the robot dog

system. Each signal command corresponded to a specific action: "Flips laterally" triggered a side roll, "Stand up" moved the robot from a prone position to standing on all fours, "Climb stairs" initiated stair-climbing mode, and "Dance" initiated a dance sequence. During control, the operator's spoken instructions were captured and recorded by SAAS, then transmitted to the robot dog system via Bluetooth. The neural network model processed the signals and determined their types. The robot's microcontroller control module translated the instructions into physical actions, successfully executing "Stand up", "Flips laterally", "Dance" and "Climb stairs" as directed (Fig. 5c). The entire process of remotely controlling the robot dog in real-time is shown in Supplementary Movie S4.

### Speech interaction involving people with speech disorders

Diseases such as Parkinson's disease, stroke, ALS and oral cancer can cause symptoms such as nerve damage or muscle hardening, which can lead to temporary or permanent loss of voice. This significantly impacts patients' lives, leading to isolation, reduced self-confidence[67,68], and even decreased life expectancy due to the lack of daily communication. Wearable acoustic sensors, such as the SAAS, provide a promising solution by capturing speech-related information from vocal organs, enabling daily interactions for individuals with speech disorders. Based on SAAS, we developed a voice recovery system that monitors incomplete vocal expressions or muscle movements and translates them into comprehensible speech. In the proof-of-concept demonstration, the speech recovery system is divided into two stages: training and testing. First, the brain of an aphasic patient generates the idea "How do you go to school?". The system begins by monitoring incomplete speech or local muscle movements, which generate one-dimensional signals with unique characteristics. This one-dimensional data is then converted into the spectrogram, resulting in two-dimensional vector data. Deep learning is applied to a large dataset of repeated experimental data to define the signal with this characteristic as an effective signal. This signal is used to express the phrase "How do you go to school?" (Fig. 6a), thereby enabling voice interaction between people with dysphonia and normal people. The vocalization information of ten daily communication sentences (sampling rate 8 kHz) was collected from one participant. The spectrogram showed that SAAS can sense the physiological mechanical sound signals produced by the vocal organs of people with disabilities. The temporal spectrogram is converted into an acoustic signal through a time domain diagram (Fig. 6b). We used pre-trained residual neural networks as a starting point for transfer learning. The one-dimensional time series-signals collected and based on the acoustic sensor are converted into two-dimensional time-frequency images through short-time Fourier transform for processing by the residual neural network. The residual neural network used in this study consisted of 53 convolutional layers and one fully connected layer. During training, feature vectors were iteratively processed from the convolutional layer to the pooling and activation layers to reduce the dimensionality of the feature space. By evaluating the consistency of the fully connected layer's softmax function, the extracted features were classified into specific targets. For the training model, data from one subject reading ten types of sentences were used as the input sequence for classification. During preprocessing, intervals between two sentences were identified based on amplitude thresholds, and the 10-second data was segmented into discrete statements. The time-frequency diagrams of these segments were used as network inputs. The dataset was split into a training set (70%) and a validation set (30%) for model evaluation. After training, the accuracy of all states in the training and validation data tends to peak at 20 iterations (Fig. 6d), at which time the corresponding normalized loss approaches 0 (Supplementary Fig. S21). The classification results of the network after training on the training set and validation set data correspond to the actual classification, forming a confusion matrix (Fig. 6c). Principal components visualized through t-distributed stochastic neighborhood embedding (T-SNE) showed initial state (Fig. 6e) and clear clustering (Fig. 6f) of feature vectors in machine learning space. The convolutional neural network achieved an overall prediction accuracy of 99.8% for the ten studied states on both the training and validation sets. These results underscore the system's potential to facilitate speech interaction for individuals with speech disorders, enabling effective communication and enhancing their quality of life.

## Discussion

We proposed a wireless SAAS based on PMUT and flexible electronics, capable of accurately characterizing the movement signals of human vocal organs in noisy environments or when wearing a mask. The sensor features a PMUT array comprising 19 micro-elements. With external signal processing circuit, the SAAS achieves a sensitivity of −198 dB and exhibits a flat frequency response within a bandwidth of 10 Hz–20 kHz, with a flatness of ±0.5 dB. A comparison of the response frequency of SAAS with previously reported sensors is provided in Supplementary Fig. S22. The sensor's excellent performance results from its high-quality materials, optimized structure, and higher resonant frequency. SAAS significantly outperforms commercial reference microphones in noisy environments (85 dB) and scenarios involving mask-wearing. Furthermore, we validated the robustness of SAAS by testing its performance across different attachment locations and head motions. The wearable device demonstrates superior mechanical and electrical properties, maintaining high consistency during motion and surpassing the performance of previously reported sensors of its kind (Supplementary Table S3). Given its robust performance, SAAS shows great potential to facilitate communication in challenging acoustic environments, such as during mask-wearing in infectious disease isolation or firefighting scenarios.

The SAAS effectively collects physiological mechanical sound signals generated by the vibration of vocal organs in harsh acoustic environments, enabling robust speech recognition across various scenarios. Integrated with machine learning, SAAS has demonstrated its ability to distinguish phonemes, tones, and words with similar pronunciations. Additionally, SAAS has successfully performed phrase recognition tasks for identity verification, achieving 100% accuracy in classifying the "Hello World" phrases of six participants. Due to its proximity to the skin and resistance to environmental noise, SAAS has proven effective in special HMI scenarios. Successfully implemented the control functionality of SAAS in both Pac-Man games and robot dog HMI systems, expanding its applicability to specific environments and user groups. For example, firefighters can control robot dogs to perform rescue tasks in hazardous fire scenes, while individuals in noisy environments can transmit specific signals in virtual reality applications. Furthermore, a participant wearing the SAAS engaged in voice interaction with another participant. After collecting and recording a significant number of biological signals, deep learning rapidly adapted speech recognition technology for a new user. This enabled the participant to articulate 10 common sentences with a 99.8% accuracy rate, demonstrating the potential of SAAS for facilitating voice interaction among individuals with speech disorders.

Future research will focus on improving several aspects of the SAAS to enhance its performance and expand its applications: advance thin-film materials with high piezoelectric coefficients and low dielectric constants, as well as high sensitivity structural designs, to improve the sensor's SNR. Optimize MEMS-based acoustic sensor processing techniques to further reduce the device's thickness, enabling greater flexibility and improved wearability. Develop application-specific integrated circuits (ASICs) to enhance system integration and minimize the impact of parasitic capacitance, thereby effectively increasing the SNR. Quantify the contribution of vocal

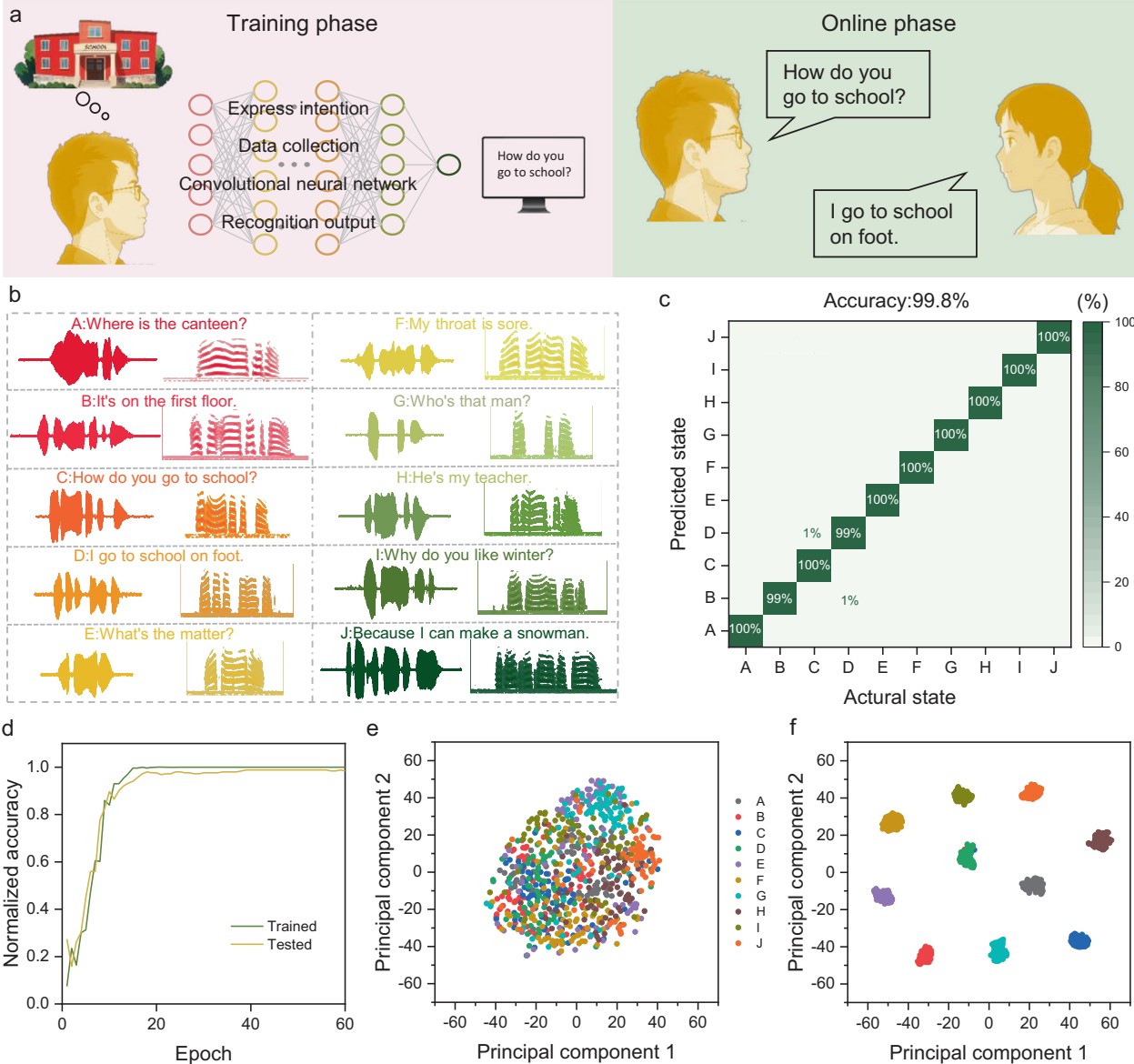

**Fig. 6 | Application of SAAS-based speech recognition system in human-to-human interaction. a** Schematic diagram of the interactive system realized through data processing, classification, and real-time display. **b** Waveforms and corresponding spectrograms of 10 sentence samples classified from daily conversations collected from participants using convolutional neural networks. **c** Confusion matrix of sentence recognition task. **d** Normalized accuracy of training and test data during 60 epoch iterations. **e** Eigenvector matrix after 60 iterations of T-SNE algorithm processing. **f** Eigenvector matrix after 60 iterations of T-SNE algorithm processing.

organs and skin movement to the signal, enabling the design of sensors better suited for comprehensive signal acquisition. Enhanced signal quality will facilitate the transition of this technology into practical applications. Overall, with its interference-resistant design, SAAS represents a promising candidate for future HMI. Further optimization could position it as a transformative technology in areas such as smart home control systems, encrypted security, and voice interaction for individuals with dysphonia. Highly integrated wearable, flexible sensors are also expected to play critical roles in these applications.

## Methods
### Ethics declaration
Every experiment human participants have been carried out following a protocol approved by an ethical commission.

Every participant gave informed written consent before being enrolled in the experiments. Participants appearing in figures and videos gave consent to the publication of content in which they may be recognizable. This includes main figures, Supplementary Figs., figures in the point-by-point response to the reviewers, and videos.

### Device finite element and design
The acoustic sensor was analyzed by finite element method using commercial software COMSOL Multiphysics 6.0, and the vibration characteristics and frequency response characteristics of acoustic sensors with different piezoelectric materials and different sensitive structures were studied. The corresponding structural dimensions were set so that the resonance points of square, hexagonal and circular acoustic sensors were unified to 300 kHz. In the finite element analysis, the material properties were configured according to the material type. According to the material properties of PZT, AlN and ScAlN, the parameters such as Young's modulus, dielectric constant and piezoelectric coefficient inside the model were adjusted.

## Device processing

The processing process of the MEMS sensor starts with the cleaning of the silicon on insulator (SOI) wafer (6N600-1-5N, Okmetic). The SOI wafer consists of a 5 μm silicon device layer, a 1 μm buried silicon oxide layer, and a 400 μm silicon substrate. A ScAlN seed layer of about 100 nm was grown through atomic layer deposition (ALD) (NLD-4000, Nano-master) to reduce the surface roughness of subsequent film deposition, thereby improving the structure and morphology of Mo and c-axis oriented ScAlN. Physical vapor phase Deposition (PVD) (Sigma, SPTS) was then used to sequentially deposit 0.3 μm Mo, 1 μm ScAlN, and 0.3 μm Mo on the ScAlN seed layer. In the subsequent patterning step, inductively coupled plasma etching (ICP) (GSEC200, NMC) was used to form the top Mo electrode pattern. ScAlN was then etched to form the bottom electrode lead-out hole. Next, 200 nm Au was deposited by magnetron sputtering (MS150X-L, FHR) and metal leads and pads were formed by the lift-off method. Finally, deep reactive ion etching (DRIE) (Omega LPX Rapier, SPTS) was performed from the back of the SOI to release cavities and form thin film vibration structures (Supplementary Fig. S7).

## Device characterization

The characterization of the device mainly focused on the micromachined MEMS acoustic sensor, and an optical microscope (MX63, Olympus) was used to characterize the sensor surface. A scanning electron microscope (SU8600, Hitachi) was used to measure the thickness and morphology of the piezoelectric films. A probe platform (EPS150RF, Cascade Microtech) and a vector network analyzer (E5080A, Keysight) were used to perform a frequency sweep from 100 to 500 kHz to measure the impedance and phase of the acoustic sensor. The growth quality of ScAlN films was tested using an X-ray diffraction instrument (Ultima IV, Rigaku). The composition of the main elements of the functional layer of the device was tested by an energy dispersive X-ray spectrometer (X-Max 50, Oxford). Use a gold wire with a diameter of 25 μm to connect the MEMS sensor to the external FPCB through a wire binding machine (WB-91D, BEE).

## Signal processing, transmission and power circuits

The amplified and filtered analog electrical signal is converted into a digital electrical signal by the Analog-to-Digital Converter (ADC) module (ADS78223), and under the control of the Microcontroller Unit (MCU) main control module (ESP32), the data is wirelessly transmitted to the Personal Computer (PC) host computer via Bluetooth (Supplementary Fig. S2). The integrated system can operate continuously on a 60 mAh lithium battery for approximately 2 h in stand-by mode. During output transmission, it can run for about 25 min (Supplementary Fig. S23).

## Device flexible packaging

The schematic diagram of the SAAS flexible package preparation process is shown in Supplementary Fig. S24. The following is the specific process. (1) Mix the A and B components of silicone (Ecoflex-0030) at a ratio of 1:1 at room temperature; (2) Place the mixed silica gel in a vacuum box to degas for 5 min; (3) Pour the degassed silica gel mixture into the mold designed in advance, so that the entire flexible circuit board of the sensor is covered with silica gel; (4) After pouring, place the entire mold into a vacuum box and continue degassing for 5 min; (5) After defoaming, the silicone mixture can be left to stand at room temperature for 4 h to solidify, and MEMS skin attachment can be obtained.

## Data acquisition and analysis

The original ADC output of the sensor is sampled through the data acquisition device, and the collected one-dimensional time domain signal is analyzed using MATLAB. Signal processing involves a second-order bandpass filter followed by conversion to a two-dimensional time-frequency image using a short-time Fourier transform.

## Machine learning algorithms for classification and prediction

We used the pre-trained residual neural network as a starting point for transfer learning. The residual neural network contains 53 convolutional layers and 1 fully connected layer. In the training model, the processing vector is iterated from the convolution layer to the pooling layer, and then to the activation layer to achieve dimensionality reduction of the feature vector. The training parameters of the network are listed in Supplementary Table 4. All measurement data of one participant are labeled with different feature types and normalized to unify the data dimensions. Among them, part of the continuous measurement data at one time (70%) is used as training data, and the other part of the data (30%) is used as validation data. The test data in another time period is used as test data. During training, feature vectors clustered by the T-SNE algorithm help visualize the evolution of the principal components in the machine learning space.

## Statistics and reproducibility

No statistical method was used to predetermine the sample size. No data were excluded from the analyses. The experiments were not randomized. The investigators were not blinded to allocation during experiments and outcome assessment.

## Reporting summary

Further information on research design is available in the Nature Portfolio Reporting Summary linked to this article.

## Data availability

The authors declare that all data supporting the findings of this study are available within the article and its supplementary files. The data sources are available at https://doi.org/10.6084/m9.figshare.27977847.v1. Any additional requests for information can be directed to, and will be fulfilled by, the corresponding authors. Source data are provided with this paper.

## Code availability

The code and algorithm in this paper are available from https://github.com/DongxiaoLi93/ML-Assisted-Wearable-Speech-Sensing-Systems[69]. Additional codes that support the findings of this study are also available from the corresponding author D.L. or X.M. upon request.

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

## Acknowledgements

The authors would like to acknowledge the financial support provided by National Key Research and Development Program of China (Grant No. 2022YFB3205400), the National Key Research and Development Program of China (Grant No. 2021YFB2012100), the Fundamental Research Funds for the Central Universities (Grant No. 2024CDJGF-005) and the Science Fund for Distinguished Young Scholars of Chongqing (Grant No. CSTB2022 NSCQJQX0006).

## Author contributions

T.L. and D.L. conceived the idea and designed all experiments. T.L., M.Z., Z.L., H.D., P.W., J.Y., W.Z. and D.L. conducted the experiments. T.L., W.Z., H.D. and Z.L. analyzed the data. T.L. wrote the manuscript. D.L., P.W. and X.M. revised the original draft. D.L. and X.M. supervised the study. All the authors discussed the results and revised the final manuscript. All the authors have approved the final version of the paper.

## Competing interests

The authors declare no competing interests.
