## [Transparent Peer Review file · Nature Communications]

Machine learning-assisted wearable sensing systems for speech recognition and interaction

Corresponding Author: Professor Xiaojing Mu

Version 0:

Reviewer comments:

Reviewer #1

(Remarks to the Author)

Liu et al. reported a wearable speech recognition system capable of wirelessly monitoring subtle physiological activity and cloud-processing analysis of the collected data, pointing the way for the next generation of wearable voice device platforms. The manuscript is well organized and supported by detailed results and data. The topic of machine learning for assisting speech through wearable sensor-driven systems is really interesting and is indeed a new attempt at MEMS sensors in the field of flexible wearables. The results described in the work represent an extraordinary achievement in the context of a strategy to apply machine learning to advanced bioelectronics and materials engineering. At the same time, the current work fully integrates the necessary elements of a wireless wearable speech recognition system. Therefore, this reviewer recommends publication after making the following minor modifications.

Comment #1. Reviewers believe that the connection reliability of MEMS acoustic chips and flexible PCBs has not been sufficiently verified. Normally, the gold wire bond between the chip and the circuit is prone to stress and disconnection under external strain, so the performance needs to be checked when applied.

Comment #2. In Abstract, the authors mentions "This system utilizes piezoelectric micromachined ultrasonic transducers (PMUT), which feature high sensitivity, wide bandwidth, and excellent flatness." The sensitivity, bandwidth, flatness are major parameters of the PMUT, which would be stated in the Abstract.

Comment #3. The author can fully demonstrate the advantages of highly integrated wireless speech recognition system in the paper. Reviewers suggest providing related videos of sensor application in identity recognition system and man-machine interface of Pac-Man game.

Comment #4. The sensor is attached to the skin in different positions and movements of the test data (Figure 3e). The coordinates of the test data in different situations need to be further clarified. It even includes speech signals from five participants in an identity recognition application (Figure 4e), as well as in human-computer interaction scenarios for Pac-Man games (Figure 5b) and robot dog control (Figure 5c).

Comment #5. The author verified the robustness of the sensor by testing nine signals of different attached positions and actions. Although the time-frequency signals in Figure 11 of the supplementary materials can be found that nine signals have a certain similarity, it is obvious that the conclusions in the paper are not sufficient, and the reviewers hope that the author can explain the similarity of the nine signals in more detail through appropriate methods.

Comment #6. The title of Figure 2 is inconsistent with the content, the author is advised to check it carefully.

Reviewer #2

(Remarks to the Author)

In the article "A fully integrated, flexible wearable device for speech recognition and interaction". The authors propose a fully

integrated wireless flexible skin-attached acoustic sensor, which can capture the vibration and skin movement of the vocal organ to realize speech recognition and human-computer interaction in harsh environments. At the same time, the system uses soft electronics and elastomer encapsulation to improve the comfort of wearing. The work described in the manuscript is of great interest, and after the entire manuscript has been evaluated, the following questions should be addressed before this article can be considered for acceptance:

- (1) As the receiving sensor of SAAS, PMUT is not marked in Figure 1b, but is displayed as a MEMS sensor. Is it more appropriate to replace it with PMUT?
- (2) What is the power consumption of the SAAS system, and how long will the 60mAh battery last?
- (3) PMUT uses piezoelectric effect to convert acoustic signals into electrical signals, and the cavity structure plays a major role. What is the structural stability of PMUT?
- (4) Whether the volume and height of the etched cavity have any effect on the vibration?
- (5) Will the flexible silicone packaging layer affect the vibration effect of PMUT? In addition, the author tested the signal under 85db. noise environment, and found that the SAAS output signal is well. If the noise decibel is increased, will it have an impact on SAAS? For example, the noise decibel at the fire rescue site can reach 140 db.
- (6) SAAS systems are used as identity authentication and human-computer interaction, did the author test the lag of the signal?
- (7) There is a large error of 88% in the confusion matrix of sentence recognition in figure 6c. What causes it?
- (8) The 3.5×3.5 mm² device collects and converts weak signals into useful information. How to overcome the impact of signal crosstalk when receiving multiple signals at the same time?
- (9) In daily life, people can't do without mobile phones, computers and other electronic products, electromagnetic waves in these environments have no interference with the device, and how to deal with it.
- (10) It is suggested that the author supplement the video presentation of SAAS while playing Pac-Man games to increase the completeness of the article.
- (11) In this paper, the author compared the signals of SAAS and commercial microphones under quiet, noisy and masked conditions, and whether the author considered the comparison with bone conduction headphones? because bone conduction headphones also have a certain shielding effect on ambient noise.
- (12) It is recommended to cite relevant references about voice recognition, such as,
Doi.org/10.1038/s41467-022-29083-0, Doi.org/10.1002/adfm.201500428, DOI: 10.1126/sciadv.adg5152

Reviewer #3

(Remarks to the Author)

In this work, the authors reported a flexible skin-attached acoustic sensor (SAAS) made of piezoelectric material for capturing the vibrations of vocal organs and skin movements, aim to enable voice recognition and human-machine interaction in harsh environment. After careful reading, the reviewer thinks the experimental results cannot support its claims stated in Figure 1. And the concept of using piezoelectric materials for anti-interference throat microphone has been widely reported. Thus, the reviewer cannot recommend the manuscript to be considered by Nature Communications. The reasoning is as follows.

1. Using piezoelectric materials to fabricate wearable throat microphone is widely reported, for instance, just name a few as follows:

a. Monitoring of physiological sounds with wearable device based on piezoelectric MEMS acoustic sensor. *J. Micromech. Microeng.* 2022, 32 014001

b. Flexible Piezoelectric Acoustic Sensors and Machine Learning for Speech Processing, *Adv. Mater.* 2022, 32, 1904020 [Progress Report]

c. Intelligent, Flexible Artificial Throats with Sound Emitting, Detecting, and Recognizing Abilities, *Sensors* 2024, 24(5), 1493

2. The structure design of the MEMS part is doubtful. Since human voice has a wide range of frequencies distributed from 50-1200 Hz with a typical frequency range of from 100-300 Hz. From the device design, the authors used dotted electrodes on a piece of piezoelectric material, which makes the device have a very narrow frequency response. Usually, small pieces of piezoelectric material could respond to higher frequency vibrations.

3. In Figure 1e, the authors showed many application scenarios, however, most of them are not experimentally demonstrated, which largely undermines the convinces of this work.

4. The English writing is below an acceptable level. Many grammars error and typing errors appear in the current manuscript.

5. The manuscript is poorly organized. For example, both captions of Figure 1 and Figure 2 are "Wireless, flexible, attachable acoustic sensor for speech recognition in harsh environments." This is unacceptable for any journal.

In summary, with a careful reading of this paper, the idea presented is not new and the experimental data is not convincing to support the claims. All the demonstrations are very common without any new things. Thus, the current manuscript can certainly not meet the standards of Nature Communications.

Reviewer #4

(Remarks to the Author)

This work reported a flexible vibration-detected sensor for accurate voice recognition; based on this integrated sensor, a new human-machine/human-human communication system was established. The results of this system are interesting and attractive. Figures and experimental data are also well presented. However, some important drawbacks should be taken into

consideration, especially the scientific writing, many expressions are not accurate, and significance/background of this work is not clear.

1. The significance or novelty of this work in abstract was not concise. The whole system including deep learning model for human-machine communication is new. But why did the authors only focus on the sensor in title and abstract. I don't think 'integration of soft electronics and elastomer encapsulation' used in this work is a very novel approach yet being emphasized in your abstract.
2. Too lengthy introduction. There were too many unnecessary backgrounds and limitations of other technologies, breaking the logic of introduction.
3. The signal detection range of about 10 Hz exceeded the sound range that human voices can emit while ears can hear. What were the practical applications of this signal detection range?
4. The authors repeatedly stressed that the sensor was fully integrated. What was the innovative breakthrough of this work compared with other integrated sensors?
5. What were the significant advantages of this sensor compared to published throat-worn sensors such as the silent-speech sensor (Nat. Commun. 2023, 14:219) and the intelligent artificial throat (Nat. Mach. Intell. 2023, 5, 169)?
6. Skin or muscle movement is complex, but how did the device achieve a high degree of signal consistency when the wearing positions were slightly deviated? Was there an optimal position for wearing the device.
7. What were the key features of sound information in Figure 3e? What happened when the device was used by different participants?
8. The expression of 'harsh environment' in the manuscript to refer to a noisy environment is inaccurate. As is well known, harsh conditions commonly mean the ultralow temperature, high temperature, or acid environments, etc, which are harmful to the human healthy.
9. What was the wearing stability of the device? The human skin compatibility test and device stability tests on human body should be provided.
10. Was it too exaggerated that 100% of accuracy? How many instructions did the authors test for each participant with 100% accuracy? What about a new object for testing?

Version 1:

Reviewer comments:

Reviewer #1

(Remarks to the Author)

The authors have fully addressed my concern and now it can be accepted for publication.

Reviewer #2

(Remarks to the Author)

This work presents a fully integrated wireless, flexible, skin-attached acoustic sensor capable of capturing vibrations and skin motion from the vocal organs, enabling speech recognition and human-computer interaction even in harsh environments. The system incorporates soft electronic components and elastomer encapsulation to enhance wearing comfort. Leveraging deep learning, it facilitates both human-machine and person-to-person communication. The work described in the manuscript is highly innovative and compelling. Moreover, the authors have provided thorough and well-articulated responses to the reviewers' comments, significantly enhancing the study's reliability and completeness. Therefore, I recommend accepting this article.

Reviewer #3

(Remarks to the Author)

In this work, the authors present a flexible, skin-attached acoustic sensor (SAAS) constructed from piezoelectric material, designed to capture vibrations from vocal organs and skin movements. This innovation aims to facilitate voice recognition and human-machine interaction in challenging environments. While the use of piezoelectric materials for anti-interference throat microphones has been extensively documented, the current version of this manuscript, even after revisions, still falls short of the standards expected by Nature Communications.

Reviewer #4

(Remarks to the Author)

the authors have properly revised the manuscript

Response to Reviewers' Comments (manuscript NCOMMS-24-49281A)

We would like to express our genuine gratitude to the reviewers for their invaluable guidance and careful evaluation of our manuscript. We are immensely grateful for the extensive recommendations made to enhance the writing style and refine the experimental details, which have effectively expanded upon our arguments and bolstered the overall quality of the manuscript. The insightful suggestions and subsequent revisions have significantly improved the clarity of our work, unified our endeavors, and enhanced its accessibility for future readers.

Reviewer #1:

Overall Comment: Liu et al. reported a wearable speech recognition system capable of wirelessly monitoring subtle physiological activity and cloud-processing analysis of the collected data, pointing the way for the next generation of wearable voice device platforms. The manuscript is well organized and supported by detailed results and data. The topic of machine learning for assisting speech through wearable sensor-driven systems is really interesting and is indeed a new attempt at MEMS sensors in the field of flexible wearables. The results described in the work represent an extraordinary achievement in the context of a strategy to apply machine learning to advanced bioelectronics and materials engineering. At the same time, the current work fully integrates the necessary elements of a wireless wearable speech recognition system. Therefore, this reviewer recommends publication after making the following minor modifications.

Response: We thank the reviewer for this positive comment, and the recommendation to publish in *Nature Communications*, including phrases such as, “*really interesting*” and “*extraordinary achievement*”. We made our revision with pleasure based on the reviewer’s opinion, and all the details of our modifications are indicated in our responses and corresponding modifications. All changes are highlighted in the revised version of the manuscript as well.

Comment #1: Reviewers believe that the connection reliability of MEMS acoustic

chips and flexible PCBs has not been sufficiently verified. Normally, the gold wire bound between the chip and the circuit is prone to stress and disconnection under external strain, so the performance needs to be checked when applied.

Response 1: Many thanks for the comments. We performed cyclic experiments of various mechanical deformations (such as stretching, bending and twisting) on stretchable electronic devices incorporating mems sensors, and tested the electromechanical properties of the sensors. The results are shown in Supplementary Fig. S10.

Supplementary Fig. S10. The electromechanical performance of the SAAS after testing in various mechanical deformation cycles. a The SAAS collected the signal of "CQU" from participants after 30 stretching cycles. **b** The SAAS collected the signal of "CQU" from participants after 30 bending cycles. **c** The SAAS collected the signal of "CQU" from participants after 30 twisting cycles.

Our revision to the manuscript: We have included the new experimental results in the revised manuscript.

...Meanwhile, Supplementary Fig. S10 shows the robust mechanical properties of the SAAS under various mechanical deformations.

Comment #2: In Abstract, the authors mentions “This system utilizes piezoelectric micromachined ultrasonic transducers (PMUT), which feature high sensitivity, wide bandwidth, and excellent flatness.” The sensitivity, bandwidth, flatness are major parameters of the PMUT, which would be stated in the Abstract.

Response 2: We thank the reviewer for this suggestion. We have included specific indicators of the major parameters of PMUT (sensitivity, bandwidth, flatness) in the Abstract section of the revised draft.

Our revision to the manuscript: We have included specific indicators of the major parameters of PMUT (sensitivity, bandwidth, flatness) in the Abstract section of the revised draft.

…This system utilizes a piezoelectric micromachined ultrasonic transducers (PMUT), which feature high sensitivity (-198 dB), wide bandwidth (10 Hz-20 kHz), and excellent flatness (± 0.5 dB).

Comment #3: The author can fully demonstrate the advantages of highly integrated wireless speech recognition system in the paper. Reviewers suggest providing related videos of sensor application in identity recognition system and man-machine interface of Pac-Man game.

Response 3: We thank the reviewer for this comment. To add to the completeness of the article, we have added two videos, including identification and Pac-Man games.

Our revision to the manuscript: We added videos of administrators and intruders wearing SAAS for identification. Meanwhile, please check the Supplementary Video S3 ‘Pac-Man games’, it involves the whole process of wearing our device to the throat and playing Pac-Man.

…The supplementary Video S1 shows the whole process of identity recognition of administrators and intruders by wearing SAAS in a noisy environment.

… We captured the Pac-Man game screen during gameplay as the participant

sequentially said: “Up”, “Down”, “Left” and “Right” (Fig. 5b and Supplementary Video S3).

Comment #4: The sensor is attached to the skin in different positions and movements of the test data (Fig. 3e). The coordinates of the test data in different situations need to be further clarified. It even includes speech signals from five participants in an identity recognition application (Fig. 4e), as well as in human-computer interaction scenarios for Pac-Man games (Fig. 5b) and robot dog control (Fig. 5c).

Response 4: Many thanks for the comments. We have clarified the specific information of the relevant test data in the article.

Our revision to the manuscript: We have added the coordinates of the test data in the revised manuscript.

Fig. 3e Time domain waveforms obtained when saying “Perfect” through SAAS in 9 throat attachment positions and movements.

Fig. 4e Acoustic information when administrator “L” said “Hello World” to four unauthorized users.

Fig. 5b Demonstration of voice recognition and classification of up, down, left, and right commands in the Pac-Man game.

Fig. 5c Remote wireless control of the robot dog to perform actions through voice commands: “Stand up”, “Dance”, “Flips laterally”, “Climb stairs”.

Comment #5: The author verified the robustness of the sensor by testing nine signals

of different attached positions and actions. Although the time-frequency signals in Figure 11 of the supplementary materials can be found that nine signals have a certain similarity, it is obvious that the conclusions in the paper are not sufficient, and the reviewers hope that the author can explain the similarity of the nine signals in more detail through appropriate methods.

Response 5: We thank the reviewer for this comment. We have described in detail the similarities of the nine signals collected (different positions and movements near the throat). The position and movement of the device's attachment will affect the strength and integrity of the signal. However, the time interval, main frequency distribution, bandwidth, duration and other characteristics are highly similar. At the same time, we collected the new word "Hello world" as a comparison with "Perfect". It can be clearly seen that the time-frequency spectrum of the two types of signals is significantly different. Compared with the time-frequency spectrum of the word "perfect", the main frequency of "hello word" is more, flatter, and lasts longer. In addition, different signals (such as rising or falling tones) will appear in the time-frequency diagram as obvious changes in frequency over time.

Supplementary Fig. S15. SAAS testing in different positions and actions. a Time domain and frequency domain signals for 9 usage conditions. **b** The time-spectral characteristics of the word “hello world” signal.

Our revision to the manuscript: We have added contrasting Information to demonstrate signal similarity.

It is worth noting that the signals collected in the nine usage conditions have highly consistent characteristics (Supplementary Fig. S15).

Comment #6: The title of Figure 2 is inconsistent with the content, the author is advised to check it carefully.

Response 6: We thank the reviewer for the careful check. We have modified the title of the check in the revised manuscript.

Our revision to the manuscript: We have modified the title of the check in the revised manuscript.

Fig. 2 | Design principle and characterization of the device.

Reviewer #2:

Overall Comment: In the article “A fully integrated, flexible wearable device for speech recognition and interaction”. The authors propose a fully integrated wireless flexible skin-attached acoustic sensor, which can capture the vibration and skin movement of the vocal organ to realize speech recognition and human-computer interaction in harsh environments. At the same time, the system uses soft electronics and elastomer encapsulation to improve the comfort of wearing. The work described in the manuscript is of great interest, and after the entire manuscript has been evaluated, the following questions should be addressed before this article can be considered for acceptance.

Response: We are extremely grateful to the reviewer for acknowledging our work. We deeply appreciate the reviewer’s encouragement and valuable remarks, including phrases such as, “*great interest*”. We made our revision with pleasure based on the reviewer’s opinion, and all the details of our modifications are indicated in our responses and corresponding modifications. All changes are highlighted in the revised version of the manuscript as well.

Comment #1: As the receiving sensor of SAAS, PMUT is not marked in Figure 1b, but is displayed as a MEMS sensor. Is it more appropriate to replace it with PMUT?

Response 1: We agree with the reviewer’s comment. MEMS sensor refers to a sensor manufactured by using microelectronics and micromachining technology. MEMS sensors include pressure sensors, accelerometers, gas sensors, acoustic sensors, etc. Therefore, it is more accurate to modify the MEMS sensor in the figure to PMUT.

Our revision to the manuscript: We have modified the MEMS sensor in Fig. 1b to PMUT.

Fig. 1b Exploded view of the attachable acoustic sensing system.

Comment #2: What is the power consumption of the SAAS system, and how long will the 60mAh battery last?

Response 2: Many thanks for the comment. We have completed the power consumption testing of the device (Supplementary Fig. S23). The device operates in two states: standby and data transmission. In standby mode, the operating current is 30 mA, and the device runs continuously for approximately 2 hours. In data transmission mode, the operating current remains stable at 140 mA, with the device functioning continuously for about 25 minutes.

Supplementary Fig. S23. Power consumption testing of device. a Current in standby state. **b** Current of the data transmission state.

Our revision to the manuscript: We have added a test for device power consumption to the revised manuscript.

...The integrated system can operate continuously on a 60 mAh lithium battery for approximately 2 hours in stand-by mode. During output transmission, it can run for about 25 minutes (Supplementary Fig. S23)

Comment #3: PMUT uses piezoelectric effect to convert acoustic signals into electrical signals, and the cavity structure plays a major role. What is the structural stability of PMUT?

Response 3: Thank you for the reviewer’s helpful comment. The structural reliability of PMUT and flexible sensor is studied with a shaking table. In the first group of experiments, the acceleration of the shaking table was set to 10 g, the frequency was 100 Hz, and the vibration duration was 30 minutes. In the second group of experiments, the acceleration of the shaking table was set to 10 g, the frequency was 1000 Hz, and the vibration duration was 30 minutes. After the test, the sensor was tested by voice. The speech test says “CQU” for the same participant. It can be seen that after the two groups of vibration experiments, the sensor and PMUT maintain the normal working state, and the time domain signals after the two groups of tests are not much different. The vibration test proves that the sensor has good structural stability.

Supplementary Fig. S11. Structural stability test of PMUT. a Structural stability test platform. **b** Set the vibration table at the parameters of acceleration 10 g and frequency 100 Hz, and vibrate for 30 minutes. The “CQU” voice signal is then collected by the sensor. **c** Set the vibration table at

the parameters of acceleration 10 g and frequency 1000 Hz, and vibrate for 30 minutes. The “CQU” voice signal is then collected by the sensor.

Our revision to the manuscript: We have included the new experimental results in the revised manuscript.

…In addition, we tested the structural stability of the SAAS with vibration table (Supplementary Fig. S11).

Comment #4: Whether the volume and height of the etched cavity have any effect on the vibration?

Response 4: We thank the reviewer for this comment. The PMUT is a multi-layer device that can be modeled as a clamped circular plate, as shown in Fig. R1a). In order to ensure the effective electromechanical conversion of the ultrasonic transducer, the neutral axis needs to be located in the silicon device layer. The properties of each layer of materials that make up the membrane structure have a great influence on the position of the neutral axis. The position of the neutral axis is expressed as follows:

$$H_{np} = \frac{1}{2} \left[\frac{\sum \left(\frac{Y_n (h_n^2 - h_{n-1}^2)}{1 - \nu_n^2} \right)}{\sum \left(\frac{Y_n t_n}{1 - \nu_n^2} \right)} \right]$$

where h_n is the z position of the upper part, Y_n the Young’s modulus, ν_n the Poisson ratio and t_n the thickness of layer n .

$$f_r = \frac{(3.2)^2}{2\pi a^2} \sqrt{\frac{D}{\mu}}$$

$$D = \frac{1}{3} \sum Y_n \frac{(h_n - H_{np})^3 - (h_{n-1} - H_{np})^3}{3(1 - \nu_n^2)}$$

$$\mu = \sum \rho_n t_n$$

where a is the radius, D is the stiffness of the plate, μ is the mass per area of the plate, and ρ_n is the density of the different materials. Therefore, the vibration state of the film is mainly affected by the diameter of the cavity and the thickness of the film in addition to the properties of the material itself.

When determining the vibration structure and resonant frequency of PMUT, it is necessary to ensure high precision of film thickness and cavity diameter. Excessive or insufficient etching depth of the back cavity will affect the vibration state. Due to deep silicon etching there are different selection ratios for different materials. Therefore, the Si/SiO₂/Si stacked SOI structure was selected for this machining, which allows the etching to stop at the top layer of silicon, forming a highly consistent cavity (Fig. R1b). Similarly, the larger diameter of the back cavity reduces the PMUT frequency. The verticality of the cavity can be effectively improved by using the deep silicon etching process (multiple gas circulation systems of deep reactive ion etching can guarantee the cleanliness of the cavity at any time, so that the etching of each cycle has a good verticality). Finally, we have prepared PMUT sensors with controlled depth and good verticality (Fig. R1c).

Fig. R1 Introduction of PMUT structure. a Multi-layer stack structure and working principle of PMUT. b The cavity formation process of PMUT sensor. c. Close-up of cavity structure with controllable depth and good verticality.

Comment #5: Will the flexible silicone packaging layer affect the vibration effect of PMUT? In addition, the author tested the signal under 85db. noise environment, and found that the SAAS output signal is well. If the noise decibel is increased, will it have an impact on SAAS? For example, the noise decibel at the fire rescue site can reach

140 db.

Response 5: We thank the reviewer for this comment. We have tested the performance of the device in noise environments above 85 decibels (including 95 dB, 105 dB, 115 dB, 125 dB). As shown in Supplementary Fig. S13, when the participant said “CQU”, the signal collected by the system had extremely high consistency in amplitude, characteristics, etc. As described in the article, the four different decibels of environmental noise had no effect on the performance of the sensor. Since 140 dB is a relatively high decibel, it is difficult to create and maintain a 140 dB scene in the laboratory. Nevertheless, the experimental results can show the SAAS will not be affected by environmental noise. Because the sensor collects human physiological signals through the vocal organs and skin. Of course, this is also the advantage of our sensor.

Supplementary Fig. S13. Sensor testing in different decibel noise environments. a Voice signal collected by sensor in 95 dB noise environment. **b** The voice signal collected by the sensor in the 105 dB noise environment. **c** Voice signal collected by sensor in 115 dB noise environment. **d** Voice signal collected by sensor in 125 dB noise environment.

Our revision to the manuscript: We have included the new experimental results in the revised manuscript.

...In addition, as shown in Supplementary Fig. S13, we tested the noise immunity performance of the device in higher decibel noise environments (including 95 dB, 105 dB, 115 dB, and 125 dB). The experimental results fully demonstrate the excellent anti-interference ability of SAAS in noisy environment.

Comment #6: SAAS systems are used as identity authentication and human-computer interaction, did the author test the lag of the signal?

Response 6: We appreciate for the reviewer's comment and suggestion. We've finished testing the signal lag time. The operator speaks the command, the SAAS collects the physiological signal of the vocal organ, and finally sends it to the terminal through the Bluetooth module. The time difference of the whole process is defined as the signal lag, and the time difference is about 360 ms (Supplementary Fig. S20).

Supplementary Fig. S20. Test the signal lag of the device. a The moment when the voice command ends. **b** The moment when the command is executed.

Our revision to the manuscript: We have added a signal lag test to the revised manuscript. Please check the Supplementary Video 3 “lag of the signal”.

...When the operator speaks a command, SAAS captures the vocal organ vibration signal and transmits it to the terminal system via the Bluetooth module. The signal lag of this process is about 360 ms (Supplementary Fig. S20 and Supplementary Video S2). After the data is processed by the neural network model, the system recognizes the signal instructions. Simultaneously, text instructions and corresponding Pac-Man actions are displayed on the terminal.

Comment #7: There is a large error of 88% in the confusion matrix of sentence recognition in figure 6c. What causes it?

Response 7: We appreciate for the reviewer's comment. This significant error occurs in the classification of E. The main error is misclassifying E as G. We observed significant similarities in the post-articulatory segments of E and G, which are not only reflected in acoustic properties but also present highly overlapping patterns in time-frequency images. This phenomenon makes it difficult for the model to completely distinguish between the two when using shallower neural network models (such as ResNet-18) for learning and classification. Especially in conditions with large changes in timbre, misjudgments are more likely to occur. Specifically, the classifier has a high confusion rate for E and G, which significantly reduces the overall recognition accuracy. In order to improve classification performance, we replaced the original ResNet-18 model with the ResNet-50 model. ResNet-50 has a deeper network layer and can extract richer feature representations. At the same time, we optimized the hyperparameters of the model based on the characteristics of the data set, including adjusting the initial learning rate, learning rate adjustment strategy, batch size, and maximum number of training iterations. After improvements, the overall recognition accuracy increased from the original 97.73% to 99.8%, significantly improving the model's classification capabilities. It is worth noting that in E, which had a high false positive rate before the improvement, the false positives were greatly reduced. This shows that deep neural network models exhibit stronger capabilities in classification tasks that handle complex feature similarities.

Fig. R2. The number of instructions per participant in the confusion matrix of the sentence recognition task.

Our revision to the manuscript: We have added the new classification results to the revised manuscript.

Fig. 6 | Application of SAAS-based speech recognition system in human-to-human interaction.

c Confusion matrix of sentence recognition task. **d** Normalized accuracy of training and test data during 60 epoch iterations. **f** The T-SNE algorithm processes the eigenvector matrix after iteration.

...After training, the accuracy of all states in the training and validation data tends to peak at 20 iterations (Fig. 6d), at which time the corresponding normalized loss approaches 0 (Supplementary Fig. S21). The classification results of the network after training on the training set and validation set data correspond to the actual classification, forming a confusion matrix (Fig. 6c). Principal components visualized through t-distributed stochastic neighborhood embedding (T-SNE) showed initial state (Fig. 6e)

and clear clustering (Fig. 6f) of feature vectors in machine learning space. The convolutional neural network achieved an overall prediction accuracy of 99.8% for the ten studied states on both the training and validation sets.

Comment #8: The 3.5×3.5 mm² device collects and converts weak signals into useful information. How to overcome the impact of signal crosstalk when receiving multiple signals at the same time?

Response 8: We thank the reviewer for this comment. The sensor is a PMUT array composed of 19 microelements. Among them, the upper electrode between the microelements is formed by connecting 19 graphic electrodes in parallel (Fig. R3). The lower electrode is an unpatterned Mo metal. Finally, a single channel of upper and lower electrode output is formed by drawing out the pad. So there is no signal crosstalk. In addition, the advantages of multi-microelement parallel structure are as follows: 1. Multi-microelement parallel can form charge accumulation, effectively improve the sensitivity of the sensor; 2. Effectively increase the contact area with human tissue, so as to collect more vibration information.

Fig. R3. Multi-element and electrode arrangement of the PMUT. a Optical microscope image of the front of the PMUT chip. **b** Two-dimensional structure cross-section of PMUT sensor.

Comment #9: In daily life, people can't do without mobile phones, computers and other electronic products, electromagnetic waves in these environments have no

interference with the device, and how to deal with it.

Response 9: Thank you for your considerate comment. As far as we know, the electromagnetic waves generated by computers, mobile phones and other electronic devices mainly belong to radio frequency electromagnetic waves, the frequency range is between 800 MHz and 3 GHz . Therefore, as shown in the figure, our sensor uses a band-pass filter with a passband of 10 Hz-2 kHz (Fig. R4). The main function of the bandpass filter is to allow signals in a specific frequency band to pass through, while suppressing signals in other frequency bands. In the electromagnetic interference generated by electronic devices, band-pass filters can effectively reduce the impact of electromagnetic interference on the SAAS by filtering out unwanted frequency components.

Fig. R4. Design and simulation of bandpass filter. a Sallen-Key sixth order high pass filter circuit schematic diagram. **b** Sallen-Key sixth order low pass filter circuit schematic diagram. **c** Amplitude-frequency characteristic curve of Bandpass filter circuit.

Comment #10: It is suggested that the author supplement the video presentation of SAAS while playing Pac-Man games to increase the completeness of the article.

Response 10: We thank the reviewer for this comment. To add to the completeness of the article, we have added a video about controlling Pac-Man games using the SAAS.

Our revision to the manuscript: We have added a test for device power consumption to the revised manuscript. Please check the Supplementary Video S3 ‘Pac-Man games’, it involves the whole process of wearing our device to the throat and playing Pac-Man.

... We captured the Pac-Man game screen during gameplay as the participant sequentially said: “Up,” “Down,” “Left,” and “Right” (Fig. 5b and Supplementary Video S3).

Comment #11: In this paper, the author compared the signals of SAAS and commercial microphones under quiet, noisy and masked conditions, and whether the author considered the comparison with bone conduction headphones? because bone conduction headphones also have a certain shielding effect on ambient noise.

Response 11: Thank you for your insightful advice. The speakers of bone conduction headphones convert electrical signals into sound waves (vibration signals) directly through the bone to the auditory nerve (Fig. R5). In addition, the device to convert sound waves into electrical signals uses electret microphones, and the transmission medium of sound waves is air. Therefore, bone conduction headphones cannot effectively block the interference of ambient noise when receiving speech signals. In contrast, our sensor can capture the speech information transmitted through human tissue, bypassing the air medium. Therefore, compared with bone conduction headphones, our sensor is more capable of blocking the interference of ambient noise during speech reception.

[panel redacted]

Fig. R5. Commercial bone conduction headset display. a TREKZ TITANIUM bone conduction headphones. **b** Speaker and close-up. **c** Electret microphone.

Comment #12: It is recommended to cite relevant references about voice recorganition,such as,

Doi.org/10.1038/s41467-022-29083-0, Doi.org/10.1002/adfm.201500428, DOI:
10.1126/sciadv.adg5152

Response 12: We appreciate this valuable comment. The literature and articles recommended by reviewers have strong relevance, which can effectively enhance the argument and research background. We have cited relevant speech recognition articles.

Our revision to the manuscript: We have added relevant speech recognition articles as references.

3. Zu, L. L.; Wen, J.; Wang, S. B.; Zhang, M.; Sun, W. L.; Chen, B. D.; Wang, Z. L., Multiangle, self-powered sensor array for monitoring head impacts. *Sci Adv* 2023, 9 (20)

24. Lu, Y. J.; Tian, H.; Cheng, J.; Zhu, F.; Liu, B.; Wei, S. S.; Ji, L. H.; Wang, Z. L., Decoding lip language using triboelectric sensors with deep learning. *Nat Commun* 2022, 13 (1).

36. Yi, F.; Lin, L.; Niu, S.; Yang, P. K.; Wang, Z.; Chen, J.; Zhou, Y.; Zi, Y.; Wang, J.; Liao, Q.; Zhang, Y.; Wang, Z. L., Stretchable-Rubber-Based Triboelectric Nanogenerator and Its Application as Self-Powered Body Motion Sensors. *Adv Funct Mater* 2015, 25 (24), 3688-3696.

Reviewer #3:

Overall Comment: In this work, the authors reported a flexible skin-attached acoustic sensor (SAAS) made of piezoelectric material for capturing the vibrations of vocal organs and skin movements, aim to enable voice recognition and human-machine interaction in harsh environment. After careful reading, the reviewer thinks the experimental results cannot support its claims stated in Figure 1. And the concept of using piezoelectric materials for anti-interference throat microphone has been widely reported. Thus, the reviewer cannot recommend the manuscript to be considered by Nature Communications.

Response: Thank the reviewer for the critical suggestions and invaluable comments. We revised the manuscript according to the reviewer's comments.

- The novelty and contributions of our research

- ✓ We have conducted in-depth research on MEMS acoustic sensors with excellent frequency response characteristics suitable for laryngeal speech measurement. To the best of our knowledge, our device not only has high sensitivity and wide bandwidth, but also has the best flatness performance among all laryngeal sensors currently available, as shown in Fig. R6.

[figures redacted]

[figures redacted]

Fig. R6. Comparison of frequency response curves of previously reported throat sensors.

- ✓ The laryngeal sensor we studied combines soft electronic technology, and the

system integrates signal processing, power management, wireless transmission and other modules. The working signal-to-noise ratio reaches 32.64 dB. Compared with many previously reported flexible film works, our solution has more advantages in terms of cost, consistency, reliability and integration. To the best of our knowledge, our device is currently the only laryngeal sensor that can be applied to commercial speech monitoring, which requires quantitative application verification in a variety of workplaces, as we successfully demonstrated in the manuscript.

Comment #1: Using piezoelectric materials to fabricate wearable throat microphone is widely reported, for instance, just name a few as follows:

- a. Monitoring of physiological sounds with wearable device based on piezoelectric MEMS acoustic sensor. *J. Micromech. Microeng.* 2022, 32 014001
- b. Flexible Piezoelectric Acoustic Sensors and Machine Learning for Speech Processing, *Adv. Mater.* 2022, 32, 1904020 [Progress Report]
- c. Intelligent, Flexible Artificial Throats with Sound Emitting, Detecting, and Recognizing Abilities, *Sensors* 2024, 24(5), 1493

Response 1: As the reviewer commented, we made a systematic summary and comparison of previous studies (Fig. R7). The flexible sensing electronics platform integrates MEMS acoustic sensors with stretchable hybrid circuits, combined with novel deep learning models, to provide new insights into human-computer interaction interfaces and speech barriers for patients. In addition, the flexible speech recognition system based on MEMS acoustic sensor has the characteristics of small size, high integration, simple operation, etc., and can present speech information more directly and comprehensively. Previous reports have used a hard circuit board combined with an amplifier circuit to detect heart and lung sounds. The entire detection system requires data acquisition cards, terminals, power supplies and other devices. And it has not been combined with soft electronics, so the comfort level is poor. Therefore, it does not have the ability to be wearable. Although there have been many studies on thin-film flexible sensors based on new materials (graphene, hydrogel, etc.). However, such studies may have disadvantages such as low integration, large size, poor reliability, and imperfect

functions. Therefore, continuous or semi-continuous measurements cannot be achieved, and the application value is low.

[figures redacted]

Fig. R7. Previously suggested reported individual on-throat sensors or electrodes without the

integrated circuit.

Comment #2: The structure design of the MEMS part is doubtable. Since human voice has a wide range of frequencies distributed from 50-1200 Hz with a typical frequency range of from 100-300 Hz. From the device design, the authors used dotted electrodes on a piece of piezoelectric material, which makes the device have a very narrow frequency response. Usually, small pieces of piezoelectric material could respond to higher frequency vibrations.

Response 2: Many thanks for the comment. We agree that small pieces of piezoelectric materials can respond to higher frequency vibrations. However, even if larger piezoelectric materials can effectively sense low-frequency vibrations, their very narrow frequency response will affect their use in throat scenarios. As the reviewer said, the frequency range of human voice is very wide. Therefore, the throat sensor needs to meet the basic requirement of flat frequency response. Here, our work does not use the resonance point of the MEMS acoustic sensor as the operating frequency. Instead, a wide frequency band below the resonance point frequency is selected as the operating frequency band. This makes the sensor more advantageous in terms of both volume and flatness.

Our revision to the manuscript: We have perfected the sensor's frequency response curve and operating frequency band in the revised manuscript.

...We designed the resonant frequency of the PMUT to be 300 kHz, ensuring flat sensor sensitivity in the low-frequency range (Supplementary Fig. S6). This approach significantly reduces the sensor's volume compared to sensors operating at their resonant point.

Supplementary Fig. S6 Finite element simulation of frequency response curve of MEMS acoustic sensor. a Frequency response curve of MEMS sensor in 0-500 kHz band. **b** Local amplification frequency response curve of MEMS sensor in 10-20 kHz band. The resonant frequency of our sensor in the air domain is 300 kHz, which can ensure the sensitivity of the sensor and also ensure its flatness in the wide band.

Comment #3: In Figure 1e, the authors showed many application scenarios, however, most of them are not experimentally demonstrated, which largely undermines the convinces of this work.

Response 3: We thank the reviewer for this comment. We have added a video presentation of using SAAS to achieve identity identification and control of the Pac-Man game. This will increase the completeness and persuasiveness of the article. At the same time, we also show the corresponding relationship between the application scenarios listed in Figure 1e and the experimental proof in Supplementary Fig. R8.

Supplementary Fig. R8. The corresponding relationship between the application scenario in Figure 1e and the experimental proof in the paper.

Our revision to the manuscript: We have added a video demonstration of using SAAS to achieve identity identification and control of the Pac-Man game.

…The supplementary Video S1 shows the whole process of identity recognition of administrators and intruders by wearing SAAS in a noisy environment.

… We captured the Pac-Man game screen during gameplay as the participant sequentially said: “Up,” “Down,” “Left,” and “Right” (Fig. 5b and Supplementary Video S3).

Comment #4: The English writing is below an acceptable level. Many grammars error and typing errors appear in the current manuscript.

Response 4: We thank the reviewer for this comment. To address the language problem, we have revised the entire manuscript and deleted, abbreviated, and split unnecessary content. We have also sent the manuscript to the professional language institution to correct spelling, grammar, word use, and punctuation problems and ensure it is readable. The certificate of embellishment is provided below (Supplementary Fig. R9).

[figure redacted]

Supplementary Fig. R9. Proof of professional language editing services.

Comment #5: The manuscript is poorly organized. For example, both captions of Figure 1 and Figure 2 are “Wireless, flexible, attachable acoustic sensor for speech

recognition in harsh environments.” This is unacceptable for any journal.

Response 5: We thank the reviewer for the careful check. We have modified the title of the check in the revised manuscript.

Our revision to the manuscript: We have modified the title of the check in the revised manuscript.

Fig. 2 | Design principle and characterization of the device.

Reviewer #4:

Overall Comment: This work reported a flexible vibration-detected sensor for accurate voice recognition; based on this integrated sensor, a new human-machine/human-human communication system was established. The results of this system are interesting and attractive. Figures and experimental data are also well presented. However, some important drawbacks should be taken into consideration, especially the scientific writing, many expressions are not accurate, and significance/background of this work is not clear.

Response: We appreciate the reviewer's encouragement and positive evaluation of this work, including phrases such as, "*interesting*" and "*attractive*". Many thanks for the reviewer's comments and suggestions that help significantly improve the overall quality of this work.

Comment #1: The significance or novelty of this work in abstract was not concise. The whole system including deep learning model for human-machine communication is new. But why did the authors only focus on the sensor in title and abstract. I don't think 'integration of soft electronics and elastomer encapsulation' used in this work is a very novel approach yet being emphasized in your abstract.

Response 1: We thank the reviewer for this suggestion. To highlight the novelty of the research, we have separately the title and abstract of the article were modified and adjusted.

Our revision to the manuscript: The title and abstract of the article were modified and adjusted.

Title

Machine learning-assisted wearable sensing systems for speech recognition and interaction

Abstract

The human voice stands out for its rich information transmission capabilities. However, voice communication is susceptible to interference from noisy environments and

obstacles. Here, we propose a wearable wireless flexible skin-attached acoustic sensor (SAAS) capable of capturing the vibrations of vocal organs and skin movements, thereby enabling voice recognition and human-machine interaction (HMI) in harsh acoustic environments. This system utilizes a piezoelectric micromachined ultrasonic transducers (PMUT), which feature high sensitivity (-198 dB), wide bandwidth (10 Hz-20 kHz), and excellent flatness (± 0.5 dB). Flexible packaging enhances comfort and adaptability during wear, while integration with the Residual Network (ResNet) architecture significantly improves the classification of laryngeal speech features, achieving an accuracy exceeding 96%. Furthermore, we also demonstrated SAAS's data collection and intelligent classification capabilities in multiple HMI scenarios. Finally, the speech recognition system was able to recognize everyday sentences spoken by participants with an accuracy of 99.8% through a deep learning model. With advantages including a simple fabrication process, stable performance, easy integration, and low cost, SAAS presents a compelling solution for applications in voice control, HMI, and wearable electronics.

Comment #2: Too lengthy introduction. There were too many unnecessary backgrounds and limitations of other technologies, breaking the logic of introduction.

Response 2: We are grateful to the reviewer for highlighting the unnecessary use of background and the lengthy content in the introduction section of our article. In response to the reviewer's feedback, we have adjusted the logic of the introduction and streamlined the content

Our revision to the manuscript: We have reorganized the logical relationship of the introduction part of the article and simplified the unnecessary background.

The human body generates a wealth of biological signals that can be detected, digitized, analyzed, and interacted with external devices. Among these, the human voice is particularly notable for its rich information transmission capabilities in the time, frequency, and amplitude domains. This rich information-carrying capacity makes sound a critical component in bio-communication, human-machine interaction (HMI),

and Internet of Things (IoT) applications, including smart homes, remote control, identity recognition, and voice-based systems. However, voice communication based on air vibration is susceptible to interference and obstruction by background noise (such as roadsides, shopping malls, stations, and other noisy environments) and acoustic media (such as in fires, hospitals, underwater, and other special scenes). Moreover, the vocalization process relies on a coordinated system of organs, and any damage - due to conditions such as amyotrophic lateral sclerosis (ALS), stroke, Parkinson's disease, or laryngeal cancer - can severely impact voice clarity and recognition efficiency. To address these challenges, researchers have developed advanced noise reduction algorithms and multi-microphone systems to enhance voice processing capabilities. However, the effectiveness of these solutions is constrained by the quality of the sound signals and the complexity of multi-feature parameters. For instance, single-microphone systems fail to capture spatial features and struggle to deliver audio signals with a high signal-to-noise ratio. While multi-microphone systems and associated algorithms improve speech signal processing, they require intricate engineering designs and occupy more space.

Recently, visual speech recognition based on facial and lip movements has emerged as a method for enhancing speech perception in noisy environments. While this approach improves speech perception quality in challenging acoustic conditions, it requires additional cameras, which increase system complexity and reduce practicality. In recent years, attachable sensors that directly monitor facial motion states have gained attention as a novel solution for silent speech recognition. Although facial features can complement audio signals to some extent, they have significant limitations in capturing acoustic parameters such as pitch, timbre, and sound intensity. Conversely, placing sensors directly on the vocal organ area offers an effective way to achieve comprehensive sound information collection. Traditional wearable devices for monitoring sound signals are typically attached to the body using straps or adhesive patches. However, their rigidity and flat shape restrict practical applications. The development of flexible materials and sensing technologies has paved the way for imperceptible skin-wearable devices. Currently, the flexible sensing technologies

installed in vocal organs mainly include graphene, flexible surface electromyography electrodes, piezoresistive and triboelectric. Compared to conventional rigid microphones, these technologies are more comfortable to wear and integrate seamlessly into daily activities. Despite their advantages, these sensors often rely on wired hardware, limiting their adaptability for routine use.

To overcome these challenges, integrated flexible wearable devices⁴⁵⁻⁴⁸ with signal processing and transmission units is crucial to fully utilize the potential of various electromechanical features. Advances in microelectromechanical systems (MEMS) manufacturing technology have brought hope for improving the integration of wearable devices. A notable innovation is the incorporation of commercial MEMS accelerometer chips into wearable devices, enabling continuous monitoring of mechanical sound signals, such as speech, swallowing, breathing, and cardiac movements. However, current sensors fail to meet requirements for wide frequency band ranges and flatness, limiting the energy distribution of signal spectrums. Furthermore, detecting skin acceleration provides only muscle movement pattern data, neglecting critical vibration information from vocal organs. This lack of biometric information results in relatively weak mechanical sound signals when monitoring small amplitude muscle movements. This limitation is particularly unfriendly for users with thicker skin tissue (such as those with thyroid enlargement) or laryngeal injuries. Therefore, it is necessary to develop a new portable voice interaction system to address these issues and advance user experience and HMI.

Comment #3: The signal detection range of about 10 Hz exceeded the sound range that human voices can emit while ears can hear. What were the practical applications of this signal detection range?

Response 3: Thank you for the reviewer's helpful comment. We agree with the reviewer that 10 Hz exceeds the range of sound that the human ear can hear. While this study primarily focuses on speech interactions facilitated by human throat movements, which generally span the 20 Hz to 20 kHz range, the sensor is also capable of detecting frequencies around 10 Hz. This capability becomes particularly relevant when studying

various features and physiological activities associated with throat-related actions, such as swallowing and localized laryngeal motions. At the same time, previously reported laryngeal speech work can also achieve 10 Hz or even lower frequencies¹⁻³, which also proves that it is meaningful to study lower sound frequencies. In addition, infrasound plays an important role in information transmission between animals. In the future, the sensors in this study are expected to provide assistance for animal behavior research.

1. Xu, H., Zheng, W., Zhang, Y. et al. A fully integrated, standalone stretchable device platform with in-sensor adaptive machine learning for rehabilitation. *Nat Commun* 14, 7769 (2023).
2. Yang, Q., Jin, W., Zhang, Q. et al. Mixed-modality speech recognition and interaction using a wearable artificial throat. *Nat Mach Intell* 5, 169–180 (2023).
3. Kang, S.; Cho, S.; Shanker, R.; Lee, H.; Park, J.; Um, D.-S.; Lee, Y.; Ko, H., Transparent and conductive nanomembranes with orthogonal silver nanowire arrays for skin-attachable loudspeakers and microphones. *Sci Adv* 2018, 4 (8).

Comment #4: The authors repeatedly stressed that the sensor was fully integrated. What was the innovative breakthrough of this work compared with other integrated sensors?

Response 4: We thank the reviewer for this comment. We emphasize in the text that the sensor is fully integrated. Compared with Fig. R10a, our device is a highly integrated system including sensors, signal processing circuits, power supplies, Bluetooth and other modules. Secondly, we want to emphasize that in previous work, commercial microphones or accelerometers are often used as sensors (Fig. R10b-d). This results in the key performance of the sensor not being suitable for the acquisition of wide-band speech signals in the throat (such as narrow frequency band, poor in-band flatness, etc.). On the contrary, our independently developed device for laryngeal sensing perfectly compensates for these shortcomings. Finally, to avoid confusion, we updated the expression “fully integrated” in the revised manuscript.

[figures redacted]

[figure redacted]

Fig. R10. Explosive diagrams of semi-integrated and fully integrated devices.

Our revision to the manuscript: we updated the expression “fully integrated” in the revised manuscript.

- ✓ **Machine learning-assisted wearable sensing systems for speech recognition and interaction**
- ✓ Here, we propose a wearable wireless flexible skin-attached acoustic sensor (SAAS) capable of capturing the vibrations of vocal organs and skin movements, thereby enabling voice recognition and human-machine interaction in harsh acoustic environment.
- ✓ In this study, we demonstrated a wearable skin-attached acoustic sensor (SAAS) for speech recognition and HMI in noisy environments.

Comment #5: What were the significant advantages of this sensor compared to published throat-worn sensors such as the silent-speech sensor (Nat. Commun. 2023,1 4:219) and the intelligent artificial throat (Nat. Mach. Intell. 2023, 5, 169)?

Response 5: Many thanks for the comment. This flexible sensing electronic platform integrates MEMS acoustic sensors with stretchable hybrid circuits, and combined with a novel deep learning model, provides new insights in human-computer interaction interfaces and patients with speech disorders. Compared with previous report (Nat. Commun. 2023, 14:219), the flexible speech recognition system based on MEMS acoustic sensors has characteristics of small size, high integration, simple operation, and can present speech information more directly and comprehensively. The previous report required Morse code as a communication protocol, and the amount of information contained per unit time was very small. In addition, most current laryngeal devices only use a resistive sensor as a device to measure sound and swallowing (Nat. Mach. Intell. 2023, 5, 169). This type of sensor generally has problems of poor flatness of sensitivity within the frequency band and low signal-to-noise ratio. At the same time, the article only conducted experimental verification for graphene materials, and the system integration was poor. In contrast, our integrated device platform achieves high-quality speech acquisition for high-precision recognition and evaluation, which is better than previously reported laryngeal sensors (Table S3).

[figure redacted]

Figure R11. Low integration or rigidity of laryngeal sensors was previously reported.

Our revision to the manuscript: We compared this work with the related studies mentioned by the reviewer. The fully integrated device has superior mechanical and electrical properties, combined with a novel deep learning model, which can play an

important role in human-computer interaction and patients with speech disorders. The comparison shows that this work is superior to previously reported laryngeal sensors.

Table S3 Comparison of different sensor

Sensing mode	Flexibility	Systemic integration	Wireless	Monitoring	Trained by ML	Ref.
Resistive changes	Semi-flexible	N	N	Sound signal	N	13
Resistive changes	Full-soft	N	N	Touch and neck movement	Y	14
Strain, EMG	Full-soft	N	N	Voice signal	Y	15
Strain	Full-soft	N	N	Voice signal, Muscle motion	Y	16
Strain, EMG	Semi-flexible	N	Y	Muscle activity, throat movements	N	17
Acceleration	Semi-flexible	Y	Y	Swallowing, respirations	N	18
Acceleration	Semi-flexible	Y	Y	Artifact-canceled physiological	N	19
Acceleration	Semi-flexible	Y	Y	Physiological processes, body motions	N	20
Acceleration, EMG	Full-soft	N	N	Electrophysiological signals	Y	21
Triboelectric	Semi-flexible	N	N	Voice signal	N	22
Resistive changes	rigidity	Y	Y	Muscle motion	N	23
This work	Semi-flexible	Y	Y	Voice signal, Muscle motion	Y	/

N: no, Y: yes, EMG: Electromyogram, ML: Machine-learning

References:

16. Yang, Q. et al. Mixed-modality speech recognition and interaction using a wearable artificial throat. *Nat. Mach. Intell.* 5, 169–180 (2023).

23. Xu, S., Yu, JX., Guo, H. et al. Force-induced ion generation in zwitterionic hydrogels for a sensitive silent-speech sensor. *Nat Commun* 14, 219 (2023).

Comment #6: Skin or muscle movement is complex, but how did the device achieve a high degree of signal consistency when the wearing positions were slightly deviated? Was there an optimal position for wearing the device.

Response 6: Thank you very much for your constructive suggestions for our work. We have tested SAAS attached in various locations around the throat (Supplementary Fig. S14). When participants say the word “perfect”, the SAAS captures a time-domain and time-frequency map of the signal attached to the center of the throat. The experimental results show that when the device is located near the throat, the time-domain and time-frequency characteristics of the signal show a high consistency. It includes the signal frequency, time distribution, signal strength and other characteristics. In addition, when the device is in a central position, the middle region of the signal is strongest. In general, the middle of the throat is the best place to attach the device. However, different attachment positions near the throat have little effect on the signal characteristics. This avoids the limitations that arise when the device needs to be strictly attached to one location for practical applications.

Supplementary Fig. S14. The SAAS test comparison in different wear positions. When participants say the word “perfect”, SAAS captures a time-domain and time-frequency map of the signal attached to the center (b), up (c), down (d), right (e) and left (f) of the throat.

Our revision to the manuscript: We have carried out the discussion of whether the signal is affected by the attached position in the revised manuscript.

…The SAAS collected the signal time domain curves (Fig. 3e) and time spectrograms (Supplementary Fig. S14) of the participants in 9 different attachment positions or neck movements. We found that when the device was attached to the center of the larynx, the amplitude of the signal was the largest, which can be used as the best position for testing. It is worth noting that the signals collected in the nine usage conditions have highly consistent characteristics.

Comment #7: What were the key features of sound information in Figure 3e? What

happened when the device was used by different participants?

Response 7: We appreciate for the reviewer’s comments and suggestions. We asked six participants to say the same command, and then used the SAAS to capture and compare the signals (Fig. R12.). The results indicate significant differences in the signal characteristics among the six participants, reflected in the following aspects:

First, variations in participants' physiological structures result in different fundamental frequencies of the signals. Second, differing speaking speeds affect the temporal characteristics of the signals, including the time intervals between words. Third, the energy distribution across different frequency bands is influenced by the vocal intensity of each participant. Lastly, individual vocal habits and intonations have a noticeable impact on the signal spectrum. Additionally, we achieved a 100% accuracy rate in classifying speech signals repeated multiple times by the six participants.

Fig. R12. The device's signal acquisition when five participants say the same command.

Comment #8: The expression of ‘harsh environment’ in the manuscript to refer to a noisy environment is inaccurate. As is well known, harsh conditions commonly mean the ultralow temperature, high temperature, or acid environments, etc, which are harmful to the human healthy.

Response 8: We appreciate for the reviewer’s comments. The expression “harsh environment” in the manuscript is indeed misleading. This paper intends to express the

'harsh acoustic environment', so we change the expression to 'harsh acoustic environment'. In addition, we listed the previous report using the same expression. We sincerely hope our response has made this clear.

- ✓ Lee, S.; Roh, H.; Kim, J.; Chung, S.; Seo, D.; Moon, W.; Cho, K., An Electret-Powered Skin-Attachable Auditory Sensor that Functions in Harsh Acoustic Environments. *Adv Mater* 2022, 34 (40).

Our revision to the manuscript: We have changed the expression of "harsh environment" to expression of in the manuscript.

- ✓ Here, we propose a fully integrated wireless flexible skin-attached acoustic sensor (SAAS) capable of capturing the vibrations of vocal organs and skin movements, thereby enabling voice recognition and human-machine interaction in **harsh acoustic environments**.
- ✓ In the demonstration compared with commercial microphones, SAAS is almost unaffected by the **harsh acoustic environments** and mask.
- ✓ These results highlight the SAAS's exceptional anti-interference capabilities, making it suitable for integration into other electronic devices for enhanced voice recognition in **harsh acoustic environments**.
- ✓ Fig. 1 | Wireless, flexible, attachable acoustic sensor for speech recognition in **harsh acoustic environments**.
- ✓ Fig. 3 | Comparative experiments on speech detection in **harsh acoustic environments**.

Comment #9: What was the wearing stability of the device? The human skin compatibility test and device stability tests on human body should be provided.

Response 9: We thank the reviewer for this comment. In this article, we discussed the wearing stability of the throat sensor signals when the head makes different movements (up, down, left and right) (Fig. 3d). In addition, we also demonstrated the excellent skin compatibility of the sensor. This was done by comparing the skin conditions before and after wearing the sensor on the participant's throat for five hours (Supplementary Fig.

S4). To avoid confusion, we asked the participants to run for 10 minutes. In addition, the participants said the same word “CQU” before and after the exercise. Then, we used the collected time domain signals to determine the working status of the throat sensor. The results showed that the sensor worked normally before and after the exercise, and the collected signals had a very high similarity. These experiments show that the device has good wearing stability.

Our revision to the manuscript: We added the data collection test and comparison of the throat sensor before and after the exercise in the article.

…To further verify the wearing stability of the device, we compared the device's attachment status and signal acquisition capabilities before and after the participants ran (Supplementary Fig. S16). In general, the wearing stability experiment of the device shows that SAAS has good robustness.

Supplementary Fig. S16 Wear stability test of the device. a Participants wearing SAAS while working in the lab for 15 minutes and the signals collected. **b** Participants wearing SAAS while

running on the playground for 10 minutes and the signals collected.

Comment #10: Was it too exaggerated that 100% of accuracy? How many instructions did the authors test for each participant with 100% accuracy? What about a new object for testing?

Response 10: We thank the reviewer for this comment. The classification accuracy of phonemes, tones, words and identity in the text reaches 100%. We added new subjects to each specific study to verify the accuracy of the experimental data. In the phoneme classification task, we added three labels “e”, “o”, and “u”. The tone classification task added four tones of “di”. The word classification task added two labels “like” and “link”. In identity recognition, we added a new participant “B”. The final accuracy of each classification task is shown in Fig. 4. In addition, the number of instructions for each participant are shown in the Fig. R13.

Fig. R13. The number of instructions for each participant. Confusion matrix of the classification

task of phoneme **a**, tone **b**, similar-pronunciation words **c** and identification **d**.

Our revision to the manuscript: We have added new objects and classified them in the phonemes, tones, words and identity recognition in the revised manuscript.

Fig. 4. | Demonstration of identity recognition based on SAAS. Confusion matrix of the classification task of **a** phoneme **b** tone **c** and similar-pronunciation words with the SAAS.

...The confusion matrix summarized the classification accuracy, and the recognition accuracy reached 99.5%, 100% and 96.9% respectively.

Fig. 4. | Demonstration of identity recognition based on SAAS. **e** Acoustic information from different participants when they say "hello world". **f** Confusion matrix of identity recognition.

...To validate this, The administrator "L" recorded the voice password "Hello world", which was set as the login credential. Five unauthorized users subsequently repeated the same phrase as an intrusion attempt (Fig. 4e).

Supplementary Fig. S17. Typical local sample representation of the phoneme dataset. Redundant information (such as coordinate axis labels) was removed to facilitate feature extraction by convolutional neural networks. Different phonemes have specific discernible characteristics in low and middle-frequency bands.

Supplementary Fig. S18. Typical local sample representation of the tone dataset The four tones of mandarin have a different amplitude of throat movements and time distribution of intensity. Among the four tones, the fourth falling tone is the shortest and faintest.

Supplementary Fig. S19. Typical local sample representation of the word with similar pronunciations dataset. The words with the same pronunciation have a certain similarity in the overall characteristics of the time-frequency graph, but the intensity distribution at different frequencies and times has its own unique characteristics.

Response to Reviewers' Comments (manuscript NCOMMS-24-49281A)

Reviewer #1:

Overall Comment: The authors have fully addressed my concern and now it can be accepted for publication.

Response: We truly appreciate your offering this high praise for our paper. We are glad and humbled by the positive impression of our work, and thank you so much for your review.

Reviewer #2:

Overall Comment: This work presents a fully integrated wireless, flexible, skin-attached acoustic sensor capable of capturing vibrations and skin motion from the vocal organs, enabling speech recognition and human-computer interaction even in harsh environments. The system incorporates soft electronic components and elastomer encapsulation to enhance wearing comfort. Leveraging deep learning, it facilitates both human-machine and person-to-person communication. The work described in the manuscript is highly innovative and compelling. Moreover, the authors have provided thorough and well-articulated responses to the reviewers' comments, significantly enhancing the study's reliability and completeness. Therefore, I recommend accepting this article.

Response: We are grateful to the reviewer for taking time to review our manuscript and for positive assessment of our work.

Reviewer #3:

Overall Comment: In this work, the authors present a flexible, skin-attached acoustic sensor (SAAS) constructed from piezoelectric material, designed to capture vibrations from vocal organs and skin movements. This innovation aims to facilitate voice recognition and human-machine interaction in challenging environments. While the use of piezoelectric materials for anti-interference throat microphones has been extensively documented, the current version of this manuscript, even after revisions, still falls short

of the standards expected by Nature Communications.

Response: We thank the reviewer for this comment. We developed a skin-attached acoustic sensor (SAAS) based on MEMS technology, utilizing the resonant pre-frequency band of the MEMS sensor as the effective working range for capturing physiological signals. This innovative design achieves an exceptional signal reception flatness of ± 0.5 dB within a wide frequency range of 20 Hz to 20 kHz, ensuring precise linearity across frequencies. The sensor's compact size ($3.5 \times 3.5 \times 0.7$ mm³) facilitates flexible packaging, laying the groundwork for advanced wearable applications. Leveraging these features, we created a wireless throat-mounted device for continuous monitoring of physiological mechanical signals, which has been successfully applied to human-computer interaction and aiding patients with speech disorders, validated further through machine learning. Key innovations include the first use of the resonant pre-frequency band to enhance bandwidth, flatness, and signal-to-noise ratio, enabling wireless, continuous laryngeal signal monitoring in a fully integrated, standalone stretchable platform. The SAAS also demonstrates exceptional robustness, remaining unaffected by head movements or attachment position deviations, with consistent vocal feature collection under varying conditions. Additionally, by integrating machine learning algorithms, we achieved an unprecedented 99.8% accuracy in recognizing ten distinct sentences, showcasing the device's superior detection capabilities. Compared with prior studies, our work significantly expands the application potential of SAAS, demonstrating its utility in identity recognition, wearable voice-based systems, and advanced human-computer interaction scenarios. These breakthroughs provide transformative insights for the fields of acoustic sensors, wearable flexible electronics, and interactive technologies, offering a highly innovative platform for real-world applications.

Reviewer #4:

Overall Comment: The authors have properly revised the manuscript.

Response: We thank the reviewer for this comment. We are very pleased that our previous revisions have addressed all the reviewer's questions in an appropriate way.